# Feeding ecology of broadbill swordfish (*Xiphias gladius*) in the California current

Antonella Preti[1,2,3]*, Stephen M. Stohs[3], Gerard T. DiNardo[4], Camilo Saavedra[5], Ken MacKenzie[2], Leslie R. Noble[6], Catherine S. Jones[2], Graham J. Pierce[7,8]

1 Institute of Marine Studies, University of California Santa Cruz, Santa Cruz, California, United States of America, 2 Institute of Biological and Environmental Sciences, School of Biological Sciences, University of Aberdeen, Aberdeen, Scotland, United Kingdom, 3 NOAA Fisheries, Southwest Fisheries Science Center, La Jolla, California, United States of America, 4 SCS Global Services, Emeryville, California, United States of America, 5 Centro Oceanográfico de Vigo, Instituto Español de Oceanografía, Vigo, Spain, 6 Faculty of Biosciences and Aquaculture, Nord University, Bodø, Norway, 7 Instituto de Investigaciones Marinas, Vigo, Spain, 8 Oceanlab, University of Aberdeen, Newburgh, Aberdeenshire, Scotland, United Kingdom

* Antonella.Preti@noaa.gov

**Data Availability Statement:** The data underlying the results presented in the study are available from NOAA ERDDAP / California Current Trophic Database (CCTD) at the following web-address

## Abstract

The feeding ecology of broadbill swordfish (*Xiphias gladius*) in the California Current was described based on analysis of stomach contents collected by fishery observers aboard commercial drift gillnet boats from 2007 to 2014. Prey were identified to the lowest taxonomic level and diet composition was analyzed using univariate and multivariate methods. Of 299 swordfish sampled (74 to 245 cm eye-to-fork length), 292 non-empty stomachs contained remains from 60 prey taxa. Genetic analyses were used to identify prey that could not be identified visually. Diet consisted mainly of cephalopods but also included epipelagic and mesopelagic teleosts. Jumbo squid (*Dosidicus gigas*) and *Gonatopsis borealis* were the most important prey based on the geometric index of importance. Swordfish diet varied with body size, location and year. Jumbo squid, *Gonatus* spp. and Pacific hake (*Merluccius productus*) were more important for larger swordfish, reflecting the ability of larger specimens to catch large prey. Jumbo squid, *Gonatus* spp. and market squid (*Doryteuthis opalescens*) were more important in inshore waters, while *G. borealis* and Pacific hake predominated offshore. Jumbo squid was more important in 2007–2010 than in 2011–2014, with Pacific hake being the most important prey item in the latter period. Diet variation by area and year probably reflects differences in swordfish preference, prey availability, prey distribution, and prey abundance. The range expansion of jumbo squid that occurred during the first decade of this century may particularly explain their prominence in swordfish diet during 2007–2010. Some factors (swordfish size, area, time period, sea surface temperature) that may influence dietary variation in swordfish were identified. Standardizing methods could make future studies more comparable for conservation monitoring purposes.

## Introduction

Broadbill swordfish (*Xiphias gladius*, hereafter swordfish) are the most widely distributed billfish and occur worldwide in tropical, subtropical and temperate waters from around 50˚N to

(https://oceanview.pfeg.noaa.gov/erddap/search/index.html?&searchFor=SWFSC-CCTD). An associated website with additional information and resources for interested users is at this link (https://oceanview.pfeg.noaa.gov/cctd/).

**Funding:** Support for our study includes salary funding from the NOAA Fisheries' Office of Science and Technology and contract funds from the Cooperative Institute for Marine, Earth, and Atmospheric Systems. The National Observer Program within NOAA Fisheries' Office of Science and Technology carried out sample collection. While the study fits the scope of work under the coauthors' performance plans, they received no specific funding for this work. The funders had no role in study design, analysis, decision to publish, or preparation of the manuscript.

**Competing interests:** The authors have declared that no competing interests exist.

50˚S [1–3]. They co-occur in the California Current Large Marine Ecosystem (CCLME), with several other upper trophic-level predators [4, 5], filling a similar ecosystem role to other large pelagic marine species, including other billfish species, sharks, tunas and dolphins [6]. In the CCLME, swordfish are landed in both the U.S.A. and Mexico. In the U.S.A., they are the primary target of the drift gillnet (DGN) fishery that operates mainly in the U.S. waters of the Southern California Bight (SCB). Swordfish have also been targeted historically in the Southern California Bight with harpoon gear, and more recently with deep-set buoy gear that was developed as a low-bycatch method for use during daylight hours [7–9].

Swordfish are well adapted for survival in a wide range of water temperatures from 5˚C to 27˚C; however, they are generally found in areas with sea surface temperatures (SST) above 13˚C [10]. They are highly fecund and do not seem to have discrete spawning grounds or seasons [11]. Swordfish migration patterns have not been described in depth, although tag release and recapture data indicate an eastward movement from the central Pacific, north of Hawaii, towards the U.S. West Coast [4]. There is no evidence of trans-equatorial or trans-Pacific crossing [12, 13], but data suggests that SCB swordfish may exhibit a higher level of Eastern Pacific Ocean (EPO) connectivity than previously proposed [14]. Swordfish tend to concentrate near underwater features, like seamounts and banks, and near oceanographic boundaries where sharp gradients of temperature and salinity exist [1], such as convergence zones and strong thermoclines [15]. These regions are known for having a relatively high abundance of forage species [16, 17]. Swordfish aggregate along these productive thermal boundaries between cold upwelled water and warmer water masses to forage [15, 18] and do not travel far during the first year of life [19].

Further insights into foraging come from information on vertical movement patterns. Swordfish display diurnal vertical migration, diving below the deep scattering layer by day and returning to shallower depths by night. Daytime depth distribution is hence more variable, including periods of basking behavior when swordfish are visibly present at the ocean surface, compared to a narrow depth range at night when it is concentrated near the surface [20–22]. During dives, swordfish can reach depths of up to 1136 m [12], indicating a tolerance of low water temperatures (c. 5˚C).

Like other billfish, swordfish have a number of adaptations that enhance foraging ability. They use their large bill to incapacitate and kill prey [1, 23]. Though they swim relatively fast, their large size limits maneuverability [24]. Partial endothermy and large eyes enhance foraging at depth [26]. Swordfish have also evolved a specialized muscle that functions as a brain heater. This mechanism allows them to function in cold water, which is essential to a fast-swimming predator that generally hunts on the cooler side of boundaries between oceanic water masses [1, 25–27]. Endothermy also has energy costs, suggesting that swordfish may have higher energy needs than otherwise similar heterothermic species [23]. Although they can use their sword to subdue prey items for easier consumption [28], swordfish lack teeth and ingest their food whole, physically limiting the size of prey they can handle. By contrast, sharks use their sharp teeth to tear and consume very large prey piecemeal.

Southern California is a foraging ground within the CCLME where swordfish from various regions of the eastern and central north Pacific aggregate. While the CCLME is known to be an important foraging ground for swordfish during certain times of year, the feeding habits of swordfish in this region are not well documented, especially in recent years. To date, there have been two extensive studies of swordfish feedings habits in the CCLME [29, 30] both south of the Mexico border as well as a few other less comprehensive studies [31–33]. This is the first comprehensive study on broadbill swordfish trophic ecology in the waters north of the Mexican border. The novelty of the study is not only to describe swordfish diets in the CCLME in more detail using larger sample sizes over a longer time period, but also to improve

understanding of feeding ecology by investigating sources of dietary variation. A unique feature of this research is the time of the study that overlapped with a historical expansion of jumbo squid.

This study aims to expand our knowledge of swordfish feeding ecology in the CCLME by analyzing the: (1) relative importance of different prey types; and (2) dietary variation interannually, by sub-period (within years), by area, and in relation to body size. The findings of this study can serve to inform the development of alternative approaches to better manage this economically and ecologically important species. Due to the complexity of many ecosystems, there is a need for basic knowledge of trophic interactions that are critical to understand system productivity and food chain dynamics. New policy developments have increased the relevance of feeding ecology studies, as policy-makers and fisheries managers have embraced the concept of ecosystem-based fisheries management (EBFM), thus taking a more holistic approach to resource management [34, 35]. The findings of this study can inform ecosystem models with information about trophic interactions, contributing to the development of alternative approaches to better manage this economically and ecologically important species. This type of data can also be used for ecosystem modelling based on tools such as Ecopath, Atlantis and their derivatives [36–38]. Predator diet data can provide an indication of the likelihood of competition between top predators and fisheries as well as information about ecosystem health and it can be utilized for the estimation of natural mortality of a number of prey species, some of which are commercially important. Moreover, diet patterns, by year, associated with the corresponding oceanographic conditions, can offer a tool for predicting future prey abundance and feeding behaviors in similar conditions.

## Methods

### Sampling at sea

Federal fishery observers aboard DGN vessels collected swordfish stomachs during the 2007–2014 fishing seasons. The DGN vessels operate within the U.S. EEZ, primarily in the SCB from August 15 through January 31. Because the season spans two calendar years, 'year' for this study refers to the fishing season, e.g., 2007 refers to August 2007 through January 2008. Sets are conducted using 1.8 km long drift gillnets extending from roughly 12 m to 100 m below the surface. DGN boats are active at night, setting nets within one hour before sunset and hauling in within one hour after sunrise for an average net-soaking time of approximately 12 hours. Hauling can then take 4 to 6 hours. No special permits were required to collect the stomachs as they are considered commercial fisheries discards.

Stomach samples were excised at sea, the oesophageal and pyloric ends secured with plastic cinch ties, and the stomachs then bagged, labeled and frozen. Additional data recorded at sea included set and haul-back times, water depth, SST, date, location and fish size.

### Processing in the laboratory

Stomachs were thawed, tamped with absorbent paper to remove excess water, and weighed full. Contents were then removed and the empty stomach lining weighed to obtain overall contents weight. Solid material and slurry were rinsed and sorted using a series of mesh screen sieves with mesh sizes 9.5 mm, 1.4 mm, and 0.5 mm for ease of rinsing mid-sized food boluses without losing some of the smallest items, such as fish otoliths. Degree of prey digestion was estimated using a six-point scale as follows: (1) Fresh: head, body, skin and most fins intact though some individuals may be in pieces (i.e., sliced on capture); (2) Intermediate: body and most flesh intact; fins, scales and some or all cephalopod skin may be digested; (3) Intact skeleton from head to hypural plate or body/mantle/carapace intact, or easily reconstructed to

obtain standard length measurements; (4) Unmeasurable body parts only: hard parts cannot be reassembled to obtain standard measurements, but higher taxon or species group still identifiable; (5) Digested but identifiable to a higher taxonomic level (e.g., family); and (6) Fully digested unidentifiable material; slurry. Prey items were then separated, identified to the lowest possible taxonomic level using taxonomic keys [39, 40] enumerated, measured and weighed. Fish otoliths and the upper and lower squid beaks were counted in pairs when possible, with the highest count representing the minimum number present. These numbers were added to the numbers of intact prey. Partial remains comprising only large chunks (i.e., fist size or greater) or pieces of fish in digestive state 1 or 2 were considered to be the result of swordfish feeding on prey caught in the driftnet and therefore were discarded from the analysis. Weights were grouped by taxon (not individually), while lengths of all intact individuals within a taxon were measured. Weight of a taxon was the weight of the undigested and partially digested items found in the stomach and not based on back-calculations of weight at the time of ingestion from measurements of hard parts. This approach was chosen because substantial amounts of undigested food remains were found and it is commonly used in studies of fish stomach contents [41]. A consequence of this approach is that prey eaten longer ago contribute less to the weight.

Genetic analyses were used to identify diet items that could not be identified visually. Tissue samples for DNA extraction were taken from the interior of the sample to minimize cross contamination with other prey. DNA was extracted using a DNeasy blood and tissue kit (Qiagen) following the manufacture's protocols. The "Barcode" region of the mitochondrial cyctochrome c oxidase I (COI) gene was amplified by polymerase chain reaction (PCR) following [42], using their COI-3 primer set with M13 tails. No template negative controls were run for each PCR batch to monitor for potential DNA contamination of reagents. PCR products were sequenced using BigDye v 3.1 dye terminator chemistry (Life Technologies), using the sequencing primers M13F(-21) and M13R(-27) following manufacturers' protocols. Aligned and edited sequences were entered into the BOLD v4 [43] and matches greater than 98% identity to a single taxon were considered to be the correct species assignment for the prey item.

Secondary prey items (prey of prey) were discarded when found associated with the stomachs of fresh prey (e.g., euphausiids in the stomachs of Pacific hake). In other cases, the presence of secondary prey cannot be ruled out. This is a common issue in diet analysis but is generally considered to have only minor consequences for the estimated biomass of different prey categories [29, 44].

## Data analysis

Size range for prey in fresh and intermediate state of digestion was reported by species. Mean and median prey size was calculated for prey species with at least 2 specimens.

Randomized cumulative curves depicting the relationship between number of prey taxa detected and sample size (rarefaction curves) were constructed using the Vegan package [45] in R statistical software [46] to determine the extent to which the sample size characterize the diet [47–51]. For this analysis, the order in which stomach contents were analyzed was randomized 100 times and the mean (± 2 standard deviations) number of prey taxa observed was plotted against the number of stomachs examined. A curve approaching an asymptote with low variability indicates that the number of stomachs examined is sufficient to characterize the diet [47]. To complement this visual approach, a method proposed by [52] was used to assess whether the curve had reached an asymptote. Specifically, a straight line was fitted to the right-most 4 points of the species accumulation curve. If the slope did not differ significantly from zero, then the species accumulation curve was inferred to have reached an asymptote. For

constructing such cumulative prey curves, Bizzarro et al (2007) lumped prey into higher-level taxonomic categories (e.g., crustaceans, teleosts, polychaetes). By contrast, the lowest taxonomic level to which prey had been identified was used, making it much less likely that the curves would reach an asymptote and assuring that the curves gave a more reliable picture of the adequacy of sample size to fully describe diet. Prey identified to species as well as unidentified categories were all included in the analysis. In general, if the proportion of unidentified prey species in the diet is low, the rarefaction curve tends to be a good guide to how many samples are required to sufficiently characterize diet. If the proportion of unidentified species is high, confidence in the curve will be lower, but it can remain a helpful tool. A map showing where stomach samples were collected was created with the R package 'ggplot2' (version 3.3.5) [53].

The importance of each prey type was summarized using three standard Relative Measures of Prey Quantities (RMPQs): percent frequency of occurrence (%F); percent composition by number (%N); and percent composition by weight (%W) [41, 44, 54, 55]. Stomachs which were empty or contained only slurry and/or detritus were not considered when calculating percentages. Three combined dietary indices were also used to rank prey taxon importance, namely the geometric index of importance (GII) and percentage GII (%GII) [56], the index of relative importance (IRI) and percentage IRI (%IRI) [54] and the Prey-Specific IRI (%PSIRI) [57]. These are useful indices to rank prey importance since they take into account both numerical and weight-based importance to the diet. Some authors favor GII [58–60], others favor IRI [61–63] and some %PSIRI [64, 65], while some doubt the merits of all such combined indices (see [44] and references therein). Here, each method was used to examine only the ranking of prey types, because the three combined index values are not directly comparable.

The GII, in its simplified form, is calculated as:

$$GII_j = \frac{\left(\sum_{i=1}^{n} V_i\right)_j}{\sqrt{n}}$$

where $GII_j$ = index value for the $j$-th prey category, $V_i$ = the magnitude of the vector for the $i$-th RMPQ of the $j$-th prey category, and $n$ = the number of RMPQs used in the analysis (in this case 3, since %W, %N and %F were used).

The $\%GII_j$ converts $GII_j$ values to a percentage scale:

$$\%GII_j = \frac{\left(\sum_{i=1}^{n} V_i\right)_j}{n}$$

The IRI for the $j$-th prey category is calculated as:

$$IRI_j = \left(\%N_j + \%W_j\right) * \%F_j$$

The IRI value was also converted to a percentage, which is arguably more useful for comparisons among studies [66]:

$$\%IRI_j = 100\, IRI_j / \sum_{j=1}^{n} IRI_j$$

Letting $N_{ji}$ and $W_{ji}$ denote the count and weight of species $j$ in stomach $i$ and $k$ the number of stomachs in the sample, the Prey-Specific IRI is calculated:

$$\%PSIRI_j = \frac{\%F_j \times \left(\%PN_j + \%PW_j\right)}{2}$$

where $\%PN_j = \sum_{i=1}^{k} \%N_{ji}/k$ and $\%PW_j = \sum_{i=1}^{k} \%W_{ji}/k$ are prey-specific abundance for count and weight of species $j$, respectively [57].

To analyze overall variation in swordfish diet in relation to body size, fishing area (within the SCB and beyond the SCB areas) and year, samples were categorized into groups: (1) 'Small' ($< 165$ cm) and 'Large' ($\geq 165$ cm) size categories, based on eye-to-fork length (EFL), with the cut-off chosen to produce similar samples sizes for each group; (2) 'within the SCB' (east of 120° 30'W longitude) and 'beyond the SCB' (west of 120° 30'W longitude), reflecting separation between the more inshore waters in the SCB where the northward flowing California Counter Current influences nearshore oceanography and the more offshore waters affected by the California Current as it moves southward; and (3) 'Year' was assigned based on the DGN fishing season, August 15 through January 31, such that all specimens collected in a single fishing season were assigned the year of the season's start date.

Differences in diet across size-, area- and year-groups were quantified independently and their statistical significance estimated using bootstrap simulations. In each case of the six most important prey items overall, 1000 bootstrap replicates of GII values for both groups were generated (e.g., GII for jumbo squid in stomachs of (A) small and (B) large fish) and, for each replicate, it was noted whether GII was higher in the first subgroup or in the second subgroup. If the GII value in A was higher than the GII value in B in more than 95% of replicates, the species is significantly more important in the diet of group A than in the diet of group B (and vice versa). All measures were calculated using R statistical software [46]. No index value was estimated if the sample size was less than 10, since small samples are known to produce biased values [67].

To summarize relationships between diet composition in terms of the importance of different prey items (response variables) and potential explanatory factors, redundancy analysis (RDA) was used, as implemented in Brodgar 2.7.4 (www.brodgar.com). Rare prey taxa that were found in less than 4 stomachs were removed prior to this analysis. The swordfish sample comprised 289 individuals (samples with food and EFL available) and the effects of 5 explanatory variables on the diet (prey numbers (N)) were considered: area (within the SCB and beyond the SCB), year (2007, 2008–2010, 2011–2014), half-year (August 15 through November 7 and November 8 through January 31), predator size (EFL) and SST (which was available for each haul and was measured at the beginning of the set). Half-year divides each year in the study period that reflects the DGN fishing season (August 15 through January 31 of the following year) in two equal time portions. Years were grouped to reduce the number of distinct levels of the 'years' variable relative to the sample size and to retain a reasonable number of observations per year grouping. This approach concentrates more observations on each distinct level of the year variable, potentially increasing the reliability of our inferences about year. Categorical variables were replaced by "dummy" variables. That is, a variable with X categories is replaced by X-1 binary (0–1) variables, each signifying that the original categorical variable takes or does not take a particular value. In all analyses, only X-1 binary variables are entered because once the value of all these is specified the value of the last one is already known. Data were transformed using Chord distance [68–70], a method that allows assignment of a low weighting to rare prey species.

To examine the relationship between the importance of individual prey types and the various explanatory variables, Generalized Additive Modelling (GAM) was used. GAM is an extension of the regression-based statistical modelling approach that is suitable when the response variable is not (necessarily) normally distributed and there is no reason to expect linear relationships between response and explanatory variables. In linear regression, the slope values (regression coefficients) quantify the relationships between the response variable and each of the explanatory variables, while GAM uses "smoothing" functions to capture these relationships. The default smoothing function used in the GAM function in the *mgcv* package in R [71] (and also used in Brodgar statistical software) is the thin plate regression spline. The complexity of the resulting curve is normally determined by the fitting routine ("cross-validation") but can be restricted by the user, and is summarized in the "degrees of freedom", with high values indicating more complex curves. If the degrees of freedom of a smoother are equal to or close to 1, this implies an approximately linear function. When applying GAM, it is necessary to consider the distribution of the response variable, which is likely to depend on the nature of the variable studied. In this study, the data are in the form of prey counts for the main prey species. Some prey occurred in large numbers and the distribution of the number of prey per stomach is likely to be strongly right-skewed, hence a negative binomial distribution was used. The explanatory variables were the same used for RDA (continuous: EFL, year and SST; factors: area and half-year). Half-year is a stand-alone binary variable which is not nested within year. The number of knots, k, was limited to 4 to avoid overfitting in the case of explanatory variables for which relatively simple relationships would be expected, e.g., body size. The forwards selection method was used for model fitting. To avoid the model misspecification, the optimal GAM model was validated by checking for influential data points and looking for patterns in the distribution of residuals [72, 73]. GAMs were fitted using count data for all of the top seven ranked prey items (based on GII). The Akaike Information Criterion (AIC) and Deviance Explained (DE) are alternative model selection criteria for GAMs. Both AIC and DE are reported in the paper, and AIC was used for model selection. The AIC trades off higher values of the likelihood function against a penalty for adding more parameters. Because the negative of the likelihood function enters the AIC and the penalty term is positive, lower values of the AIC indicate a better model fit to the data [74]. Model selection was based on choosing the one with the lowest AIC.

## Results

### Sample composition

A total of 299 broadbill swordfish (*Xiphias gladius*) stomachs were collected during 103 observed DGN trips in the CCLME (Fig 1). Samples were collected from 2007–2014 throughout the CCLME but especially in the southeast, where the fishing is mainly concentrated. SST at the time of sample collection ranged from 14.3°C to 21.9°C (mean 17.9°C). Swordfish ranged in size from 74 to 245 cm EFL (Fig 2). DeMartini et al (2000) provided median body size at sexual maturity ($L$50) for males (102 cm ± 2.5 (95% CI) cm EFL) and females (144 ± 2.8 cm EFL). Based on these estimates, almost all the animals in this study were above the typical size at maturity for males and a majority were above the typical size at maturity for females; as noted above, sex was not determined. Of the 299 swordfish stomachs examined, 292 contained food remains belonging to 60 different prey taxa overall. Ninety-one percent of the food items were in an advanced state of digestion (stages 4 and 5). Swordfish size groups, areas and years presented different numbers of stomach samples (Table 1).

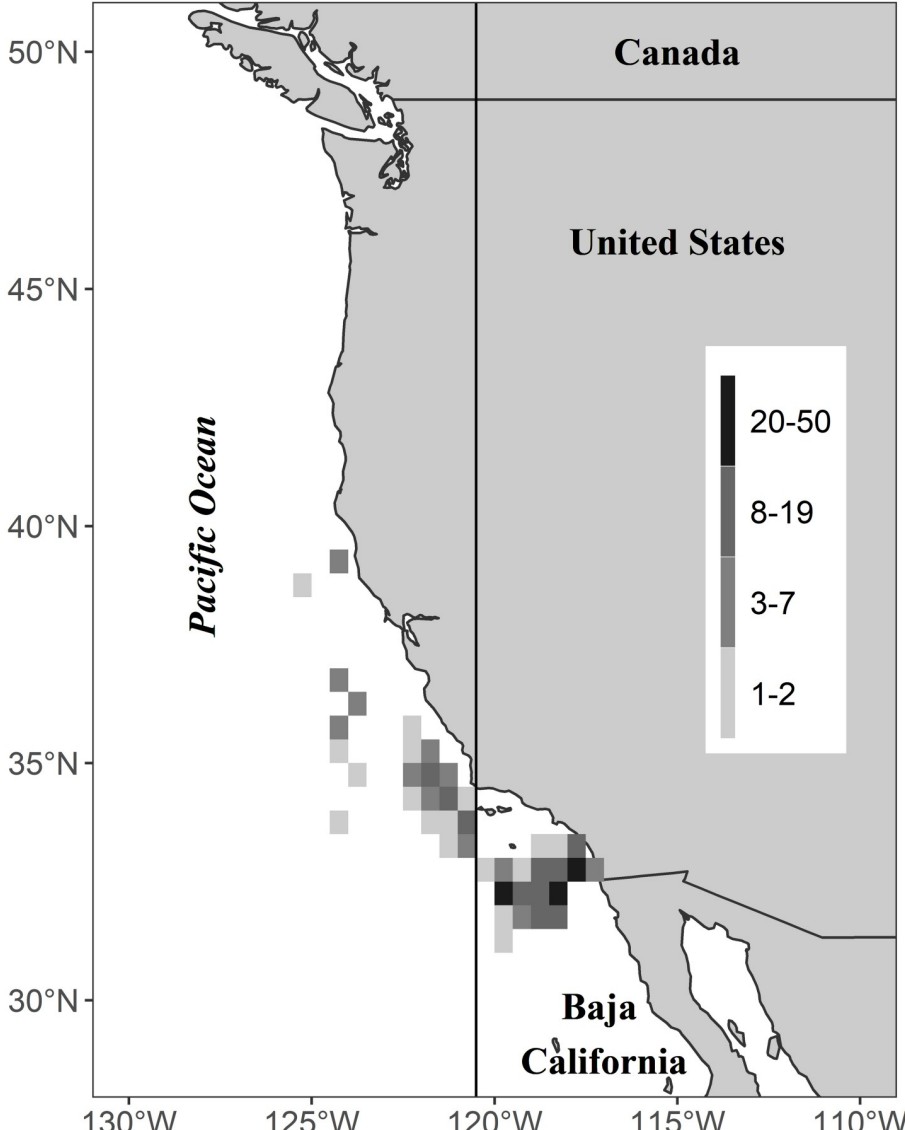

**Fig 1. Collection areas of swordfish used for diet analysis.** Number of samples (individuals) is indicated by greyscale in the legend. Map shows the northern part of the California Current Large Marine Ecosystem (CCLME) that extends to the tip of Baja California. Vertical line separates the two areas: within the Southern California Bight (SCB, east of 120º 30'W) and beyond the SCB subregion (west of 120º 30'W). The coastline was imported from the public domain Natural Earth project, via the 'maps' package [75].

Prey size was measured for 328 specimens of 22 prey species in a fresh and intermediate state of digestion. Prey size range was reported and mean and median prey size by species were calculated for prey with at least 2 specimens available (Table 2).

## Sample size sufficiency

The cumulative prey curve did not reach an asymptote for the swordfish stomachs analyzed (Fig 3). The terminal portion of the curve (4 last points) had a slope that differed significantly from zero (p = 0.0009). Nevertheless, the fact that the curve starts to asymptote indicates that the majority of prey taxa present in the diet of the swordfish (at the temporal and spatial scale of the present study) are likely to be represented in these analyses.

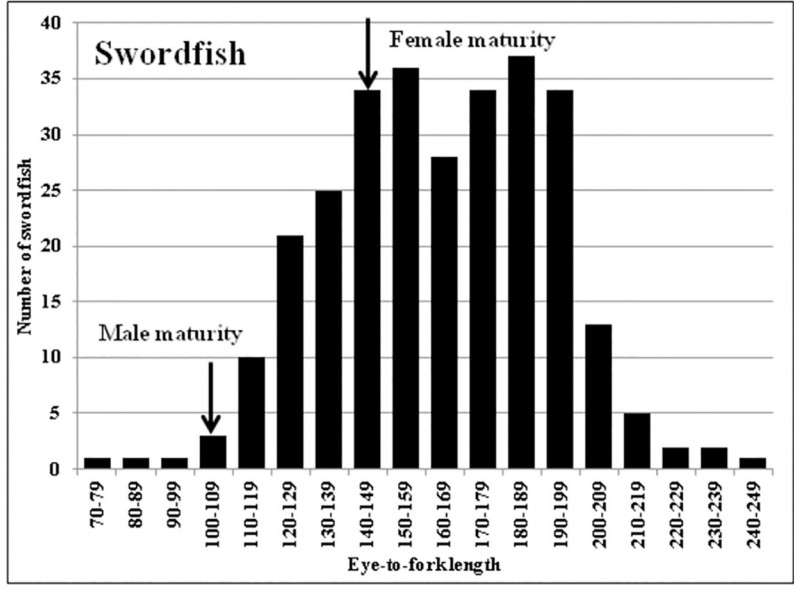

**Fig 2. Length-frequency distribution of swordfish sampled in the diet study.** *N* = 293. *Arrows* indicate typical sizes at maturity for males and females [76]. Eye-to-fork length is measured in cm. (Size was not determined for 6 individuals of the 299 sampled).

## Indices of prey importance

Table 3 lists each of the RMPQs for all prey found, as well as the calculated GII, %GII, IRI and %IRI values. Rankings of prey taxa based on GII and IRI were nearly identical. Jumbo squid (*Dosidicus gigas*) (%GII = 44.25; %IRI = 56.47; %PSIRI = 36.75) was the most important prey item by weight, number and according to the two combined indices. The boreopacific gonate squid (*Gonatopsis borealis*) (%GII = 29.08; %IRI = 20.14; %PSIRI = 12.46) was the second most important prey according to GII and IRI, and the most important by frequency of

**Table 1. Number of stomach samples by swordfish size, area and year.** "All" = number of all stomachs; "w/food" = number of stomachs with at least one prey item; "% w/food" = % of stomachs with at least one prey item.

|  | All | w/food | % w/food |
|---|---|---|---|
| **Size** |  |  |  |
| EFL < 165 cm | 149 | 148 | 99.3 |
| EFL ≥ 165 cm | 144 | 140 | 97.2 |
| **Area** |  |  |  |
| Within the SCB | 203 | 199 | 98.0 |
| Beyond the SCB | 96 | 93 | 96.9 |
| **Year** |  |  |  |
| 2007 | 48 | 47 | 97.9 |
| 2008 | 17 | 16 | 94.1 |
| 2009 | 38 | 37 | 97.4 |
| 2010 | 12 | 12 | 100 |
| 2011 | 56 | 54 | 96.4 |
| 2012 | 37 | 36 | 97.3 |
| 2013 | 57 | 56 | 98.2 |
| 2014 | 34 | 34 | 100 |

**Table 2. Size range, mean and median for 328 swordfish prey items in a fresh and intermediate state of digestion.**

| Prey name | N | Size Range | Mean | Median |
|---|---|---|---|---|
| Jumbo squid, *Dosidicus gigas* | 113 | 90–650 | 292 | 280 |
| Pacific hake, *Merluccius productus* | 76 | 180–475 | 356 | 376 |
| Boreopacific gonate squid, *Gonatopsis borealis* | 23 | 110–285 | 199 | 192 |
| Duckbill barracudina, *Magnisudis atlantica* | 21 | 225–370 | 284 | 275 |
| Pacific saury, *Cololabis saira* | 19 | 170–275 | 212 | 215 |
| Market squid, *Doryteuthis opalescens* | 15 | 90–120 | 105 | 105 |
| Pacific pomfret, *Brama japonica* | 11 | 106–380 | 270 | 270 |
| Luvar, *Luvarus imperialis* | 8 | 445–550 | 516 | 522 |
| King-of-the-salmon, *Trachipterus altivelis* | 6 | 100–360 | 246 | 285 |
| Jack mackerel, *Trachurus symmetricus* | 5 | 195–530 | 355 | 310 |
| Slender barracudina, *Lestidiops ringens* | 4 | 190–200 | 197 | 200 |
| Pacific mackerel, *Scomber japonicus* | 4 | 170–260 | 230 | 245 |
| Chubby pearleye, *Rosenblattichthys volucris* | 4 | 180–210 | 191 | 187 |
| Pacific sardine, *Sardinops sagax* | 4 | 175–245 | 208 | 206 |
| Flowervase jewell squid, *Histioteuthis dofleini* | 3 | 160–220 | 182 | 165 |
| *Nansenia* spp. | 2 | 265, 270 | 267 | 267 |
| *Onychoteuthis* sp. | 2 | 165, 270 | 217 | 217 |
| Splitnose rockfish, *Sebastes diploproa* | 2 | 290, 310 | 300 | 300 |
| Smalleye squaretail, *Tetragonurus cuvieri* | 2 | 125, 132 | 128 | 128 |
| Cock-eyed squid, *Histioteuthis heteropsis* | 2 | 150, 210 | 180 | 180 |
| Spotted barracudina, *Arctozenus risso* | 1 | 230 | | |
| Halfmoon, *Medialuna californiensis* | 1 | 210 | | |

occurrence. Other important squid prey included *Abraliopsis* sp. (%GII = 16.31; %IRI = 4.61; %PSIRI = 4.44), *Gonatus* spp. (%GII = 14.48; %IRI = 2.82; %PSIRI = 2.89) and market squid (*Doryteuthis opalescens*) (%GII = 13.66; %IRI = 4.24; %PSIRI = 5.42). Pacific hake (*Merluccius productus*) (%GII = 12.59; %IRI = 4.57; %PSIRI = 10.50) was the highest ranked teleost prey species, ranked sixth by GII. Swordfish also preyed on barracudinas (Paralepididae), several

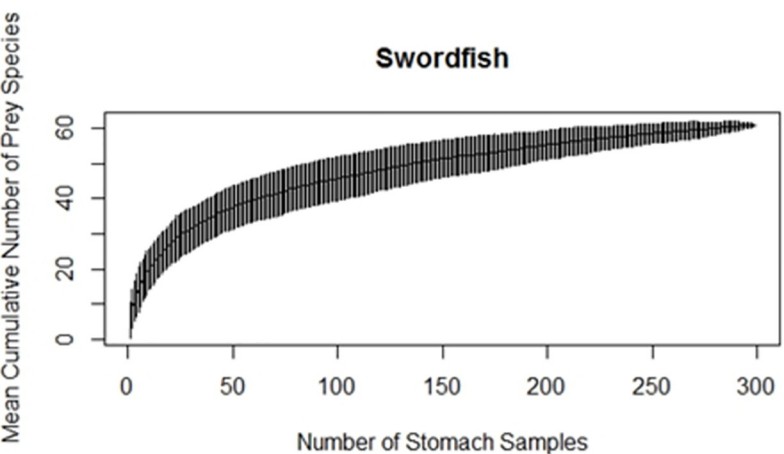

**Fig 3. Cumulative prey curve (rarefaction curve) for swordfish (prey identified at the lowest possible taxonomic level).**

**Table 3. Quantitative prey composition of the broadbill swordfish (*Xiphias gladius*) in the CCLME.** A total of 292 stomachs containing food was examined. Prey items are shown in order of decreasing GII value. W = weight (g) for the given prey taxon, %W is the same value expressed as a percentage of the total weight summed across all prey taxa; N = number of prey individuals; F = frequency of occurrence (number of stomachs in which the prey taxon occurred); %F = frequency of occurrence expressed as a percentage of the number of (non-empty) stomachs examined; GII = geometric index of importance; IRI = index of relative importance; % PSIRI = percentage prey-specific IRI.

| Prey Taxon | W (g) | %W | N | %N | F | %F | GII | %GII | IRI | %IRI | %PSIRI |
|---|---|---|---|---|---|---|---|---|---|---|---|
| **Jumbo squid, *Dosidicus gigas*** | 131892.7 | 53.27 | 1061 | 20.23 | 173 | 59.25 | 76.64 | 44.25 | 4354.96 | 56.47 | 36.75 |
| **Boreopacific gonate squid, *Gonatopsis borealis*** | 19949.8 | 8.06 | 884 | 16.86 | 182 | 62.33 | 50.37 | 29.08 | 1552.94 | 20.14 | 12.46 |
| *Abraliopsis* sp. | 45.1 | 0.02 | 464 | 8.85 | 117 | 40.07 | 28.25 | 16.31 | 355.26 | 4.61 | 4.44 |
| *Gonatus* spp. | 181.6 | 0.07 | 299 | 5.7 | 110 | 37.67 | 25.08 | 14.48 | 217.56 | 2.82 | 2.89 |
| **Market squid, *Doryteuthis opalescens*** | 1447.6 | 0.58 | 538 | 10.26 | 88 | 30.14 | 23.66 | 13.66 | 326.81 | 4.24 | 5.42 |
| **Pacific hake, *Merluccius productus*** | 36360.1 | 14.69 | 331 | 6.31 | 49 | 16.78 | 21.81 | 12.59 | 352.37 | 4.57 | 10.50 |
| **Duckbill barracudina, *Magnisudis atlantica*** | 4568.6 | 1.85 | 218 | 4.16 | 84 | 28.77 | 20.07 | 11.59 | 172.67 | 2.24 | 3.01 |
| **Unidentified Teleostei** | 2316.9 | 0.94 | 119 | 2.27 | 65 | 22.26 | 14.7 | 8.49 | 71.35 | 0.93 | 1.61 |
| **Chubby pearleye, *Rosenblattichthys volucris*** | 810.6 | 0.33 | 166 | 3.17 | 49 | 16.78 | 11.71 | 6.76 | 58.61 | 0.76 | 1.75 |
| **Jack mackerel, *Trachurus symmetricus*** | 6668.2 | 2.69 | 72 | 1.37 | 28 | 9.59 | 7.88 | 4.55 | 38.99 | 0.51 | 2.03 |
| *Nansenia* spp. | 510.9 | 0.21 | 124 | 2.36 | 32 | 10.96 | 7.81 | 4.51 | 28.17 | 0.37 | 1.29 |
| *Onychoteuthis borealijaponica* | 656.6 | 0.27 | 60 | 1.14 | 35 | 11.99 | 7.73 | 4.47 | 16.89 | 0.22 | 0.71 |
| **Slender barracudina, *Lestidiops ringens*** | 330 | 0.13 | 92 | 1.75 | 29 | 9.93 | 6.82 | 3.94 | 18.75 | 0.24 | 0.94 |
| **Pacific pomfret, *Brama japonica*** | 5241.6 | 2.12 | 41 | 0.78 | 24 | 8.22 | 6.42 | 3.71 | 23.83 | 0.31 | 1.45 |
| **Pacific sardine, *Sardinops sagax*** | 1823.1 | 0.74 | 77 | 1.47 | 26 | 8.9 | 6.41 | 3.7 | 19.63 | 0.25 | 1.11 |
| **Luvar, *Luvarus imperialis*** | 19258.5 | 7.78 | 18 | 0.34 | 7 | 2.4 | 6.07 | 3.51 | 19.47 | 0.25 | 4.06 |
| **Pacific saury, *Cololabis saira*** | 1366.8 | 0.55 | 76 | 1.45 | 21 | 7.19 | 5.31 | 3.06 | 14.39 | 0.19 | 1.00 |
| **Unidentified Scopelarchidae** | 476.9 | 0.19 | 86 | 1.64 | 20 | 6.85 | 5.01 | 2.89 | 12.55 | 0.16 | 0.92 |
| **Cock-eyed squid, *Histioteuthis heteropsis*** | 1312.2 | 0.53 | 52 | 0.99 | 18 | 6.16 | 4.44 | 2.56 | 9.38 | 0.12 | 0.76 |
| **Pacific mackerel, *Scomber japonicus*** | 2180.7 | 0.88 | 66 | 1.26 | 16 | 5.48 | 4.4 | 2.54 | 11.72 | 0.15 | 1.07 |
| **Sunbeam lampfish, *Lampadena urophaos*** | 201.9 | 0.08 | 42 | 0.8 | 18 | 6.16 | 4.07 | 2.35 | 5.44 | 0.07 | 0.44 |
| **King-of-the-salmon, *Trachipterus altivelis*** | 5577.4 | 2.25 | 25 | 0.48 | 13 | 4.45 | 3.86 | 2.39 | 10.59 | 0.16 | 1.37 |
| **Flowervase jewell squid, *Histioteuthis dofleini*** | 560.1 | 0.23 | 25 | 0.48 | 15 | 5.14 | 3.37 | 1.95 | 3.61 | 0.05 | 0.36 |
| **Unidentified Eucarida** | 5.5 | <0.01 | 154 | 2.94 | 6 | 2.05 | 2.88 | 1.67 | 6.04 | 0.08 | 1.48 |
| **Unidentified Teuthoidea** | 202 | 0.08 | 15 | 0.29 | 12 | 4.11 | 2.58 | 1.49 | 1.51 | 0.02 | 0.19 |
| **Spotted barracudina, *Arctozenus risso*** | 67.9 | 0.03 | 14 | 0.27 | 8 | 2.74 | 1.75 | 1.01 | 0.81 | 0.01 | 0.15 |
| *Histioteuthis* spp. | 56.7 | 0.02 | 9 | 0.17 | 8 | 2.74 | 1.69 | 0.98 | 0.53 | 0.01 | 0.10 |
| *Argonauta* sp. | 13.1 | 0.01 | 8 | 0.15 | 8 | 2.74 | 1.67 | 0.97 | 0.43 | 0.01 | 0.08 |
| **Striped mullet, *Mugil cephalus*** | 1737.8 | 0.7 | 8 | 0.15 | 4 | 1.37 | 1.28 | 0.74 | 1.17 | 0.02 | 0.43 |
| *Octopoteuthis* sp. | 2.1 | <0.01 | 6 | 0.11 | 6 | 2.05 | 1.25 | 0.72 | 0.24 | <0.01 | 0.06 |
| **Bigfin lampfish, *Symbolophorus californiensis*** | 5.4 | <0.01 | 7 | 0.13 | 5 | 1.71 | 1.07 | 0.62 | 0.23 | <0.01 | 0.07 |
| **Sharpchin barracudina, *Stemonosudis macrura*** | 8.8 | <0.01 | 8 | 0.15 | 4 | 1.37 | 0.88 | 0.51 | 0.21 | <0.01 | 0.08 |
| *Cranchia scabra* | 4.5 | <0.01 | 5 | 0.1 | 4 | 1.37 | 0.85 | 0.49 | 0.13 | <0.01 | 0.06 |
| **Mexican lampfish, *Triphoturus mexicanus*** | <0.1 | <0.01 | 4 | 0.08 | 4 | 1.37 | 0.83 | 0.49 | 0.1 | <0.01 | 0.05 |
| **Paralepididae, Barracudinas** | 111.3 | 0.04 | 7 | 0.13 | 3 | 2.4 | 1.49 | 0.86 | 0.43 | 0.01 | 0.09 |
| **Unidentified Euphausiidae** | 3 | <0.01 | 6 | 0.11 | 3 | 2.05 | 1.25 | 0.72 | 0.24 | <0.01 | 0.06 |
| **Robust clubhook squid, *Onykia robusta*** | 43.3 | 0.02 | 4 | 0.08 | 3 | 1.37 | 0.85 | 0.49 | 0.13 | <0.01 | 0.05 |
| **Northern anchovy, *Engraulis mordax*** | 1.6 | <0.01 | 4 | 0.08 | 3 | 1.37 | 0.84 | 0.49 | 0.11 | <0.01 | 0.05 |
| **California smoothtongue, *Leuroglossus stilbius*** | <0.1 | <0.01 | 4 | 0.08 | 3 | 1.37 | 0.83 | 0.49 | 0.1 | <0.01 | 0.05 |
| **Unidentified Tunicata** | 3.5 | <0.01 | 3 | 0.06 | 3 | 1.03 | 0.63 | 0.37 | 0.06 | <0.01 | 0.04 |
| **Smalleye squaretail, *Tetragonurus cuvieri*** | 161.9 | 0.07 | 3 | 0.06 | 2 | 1.03 | 0.66 | 0.39 | 0.13 | <0.01 | 0.07 |
| *Onychoteuthis* sp. | <0.1 | <0.01 | 4 | 0.08 | 2 | 1.37 | 0.83 | 0.49 | 0.1 | <0.01 | 0.05 |
| *Japetella* sp. | <0.1 | <0.01 | 4 | 0.08 | 2 | 1.37 | 0.83 | 0.49 | 0.1 | <0.01 | 0.05 |
| **Splitnose rockfish, *Sebastes diploproa*** | 924.2 | 0.37 | 2 | 0.04 | 1 | 0.68 | 0.63 | 0.36 | 0.28 | <0.01 | 0.21 |

*(Continued)*

**Table 3.** (Continued)

| Prey Taxon | W (g) | %W | N | %N | F | %F | GII | %GII | IRI | %IRI | %PSIRI |
|---|---|---|---|---|---|---|---|---|---|---|---|
| Northern lampfish, *Stenobrachius leucopsarus* | <0.1 | <0.01 | 2 | 0.04 | 2 | 0.68 | 0.42 | 0.24 | 0.03 | <0.01 | 0.03 |
| *Octopus rubescens* | <0.1 | <0.01 | 2 | 0.04 | 2 | 0.68 | 0.42 | 0.24 | 0.03 | <0.01 | 0.03 |
| *Chiroteuthis calyx* | <0.1 | <0.01 | 2 | 0.04 | 2 | 0.68 | 0.42 | 0.24 | 0.03 | <0.01 | 0.03 |
| Albacore, *Thunnus alalunga* | 371.6 | 0.15 | 1 | 0.02 | 1 | 0.34 | 0.3 | 0.17 | 0.06 | <0.01 | 0.09 |
| *Sebastes* spp. | 3 | <0.01 | 8 | 0.15 | 1 | 2.74 | 1.67 | 0.97 | 0.42 | 0.01 | 0.08 |
| Halfmoon, *Medialuna californiensis* | 81 | 0.03 | 1 | 0.02 | 1 | 0.34 | 0.23 | 0.13 | 0.02 | <0.01 | 0.03 |
| Dogtooth lampfish, *Ceratoscopelus townsendi* | 1.5 | <0.01 | 2 | 0.04 | 1 | 0.68 | 0.42 | 0.24 | 0.03 | <0.01 | 0.03 |
| Shortbelly rockfish, *Sebastes jordani* | 0.4 | <0.01 | 2 | 0.04 | 1 | 0.68 | 0.42 | 0.24 | 0.03 | <0.01 | 0.03 |
| *Leachia dislocata* | <0.1 | <0.01 | 2 | 0.04 | 1 | 0.68 | 0.42 | 0.24 | 0.03 | <0.01 | 0.03 |
| Pacific bonito, *Sarda chiliensis* | 25.8 | 0.01 | 1 | 0.02 | 1 | 0.34 | 0.21 | 0.12 | 0.01 | <0.01 | 0.02 |
| *Auxis* sp. | 4.7 | <0.01 | 1 | 0.02 | 1 | 0.34 | 0.21 | 0.12 | 0.01 | <0.01 | 0.02 |
| *Mastigoteuthis dentata* | <0.1 | <0.01 | 1 | 0.02 | 1 | 0.34 | 0.21 | 0.12 | 0.01 | <0.01 | 0.02 |
| *Octopus* spp. | <0.1 | <0.01 | 1 | 0.02 | 1 | 0.34 | 0.21 | 0.12 | 0.01 | <0.01 | 0.02 |
| California flashlightfish, *Protomyctophum crockeri* | <0.1 | <0.01 | 1 | 0.02 | 1 | 0.34 | 0.21 | 0.12 | 0.01 | <0.01 | 0.02 |
| California headlightfish, *Diaphus theta* | <0.1 | <0.01 | 1 | 0.02 | 1 | 0.34 | 0.21 | 0.12 | 0.01 | <0.01 | 0.02 |
| Unidentified Isopoda | <0.1 | <0.01 | 1 | 0.02 | 1 | 0.34 | 0.21 | 0.12 | 0.01 | <0.01 | 0.02 |

species of coastal pelagic fishes (jack mackerel *Trachurus symmetricus*, Pacific sardine *Sardinops sagax*, Pacific saury *Cololabis saira*, northern anchovy *Engraulis mordax*), luvar (*Luvarus imperialis*), king-of-the-salmon (*Trachipterus altivelis*), halfmoon (*Medialuna californiensis*) and seven species of the family Myctophidae (Table 3). Cuts and punctures were apparent on several of prey items.

DNA analysis allowed to identify the muscle tissue of two chubby pearleye and one luvar specimens.

In general, both large and small swordfish fed on similar prey but some differences were apparent. Based on GII results, jumbo squid was the most important prey item followed by the *G. borealis*, and *Abraliopsis* sp., in both size classes. However, northern anchovy was found only in stomachs of the small size group while luvar was eaten only by large swordfish (S1 and S2 Tables). Jumbo squid, *Gonatus* spp., and Pacific hake were significantly more important in larger swordfish than smaller swordfish (S3 Table).

A comparison of the GII results by area indicated that jumbo squid and *G. borealis* were the two most important prey of swordfish in both areas. The third ranked species were *Abraliopsis* sp. within the SCB, and Pacific hake beyond the SCB. Striped mullet (*Mugil cephalus*), northern anchovy and *Sebastes* spp. were recorded only within the SCB (S4 and S5 Tables). Jumbo squid, *Gonatus* spp. and market squid were significantly more important within the SCB than beyond the SCB, while *G. borealis* and Pacific hake were significantly more important beyond the SCB (S6 Table).

Between-year comparisons showed that jumbo squid was the first ranked prey, followed by *G. borealis*, in 2007, 2008, 2010, 2012 and 2013. The importance of jumbo squid, *G. borealis*, *Gonatus* spp., market squid and Pacific hake in the diet all varied significantly between years over the study period (S15 Table). In 2009, *G. borealis* was the most important prey followed by jumbo squid. In 2011 and 2014, Pacific hake ranked first followed by *G. borealis*. Pacific hake was not present in the samples from 2008 through 2010. *Abraliopsis* sp. was important overall (ranked third) but was not present in 2012. *Gonatus* spp. ranked fourth overall but was not present in the diet in 2011 (S7–S14 Tables). Composition (%N) of swordfish diet

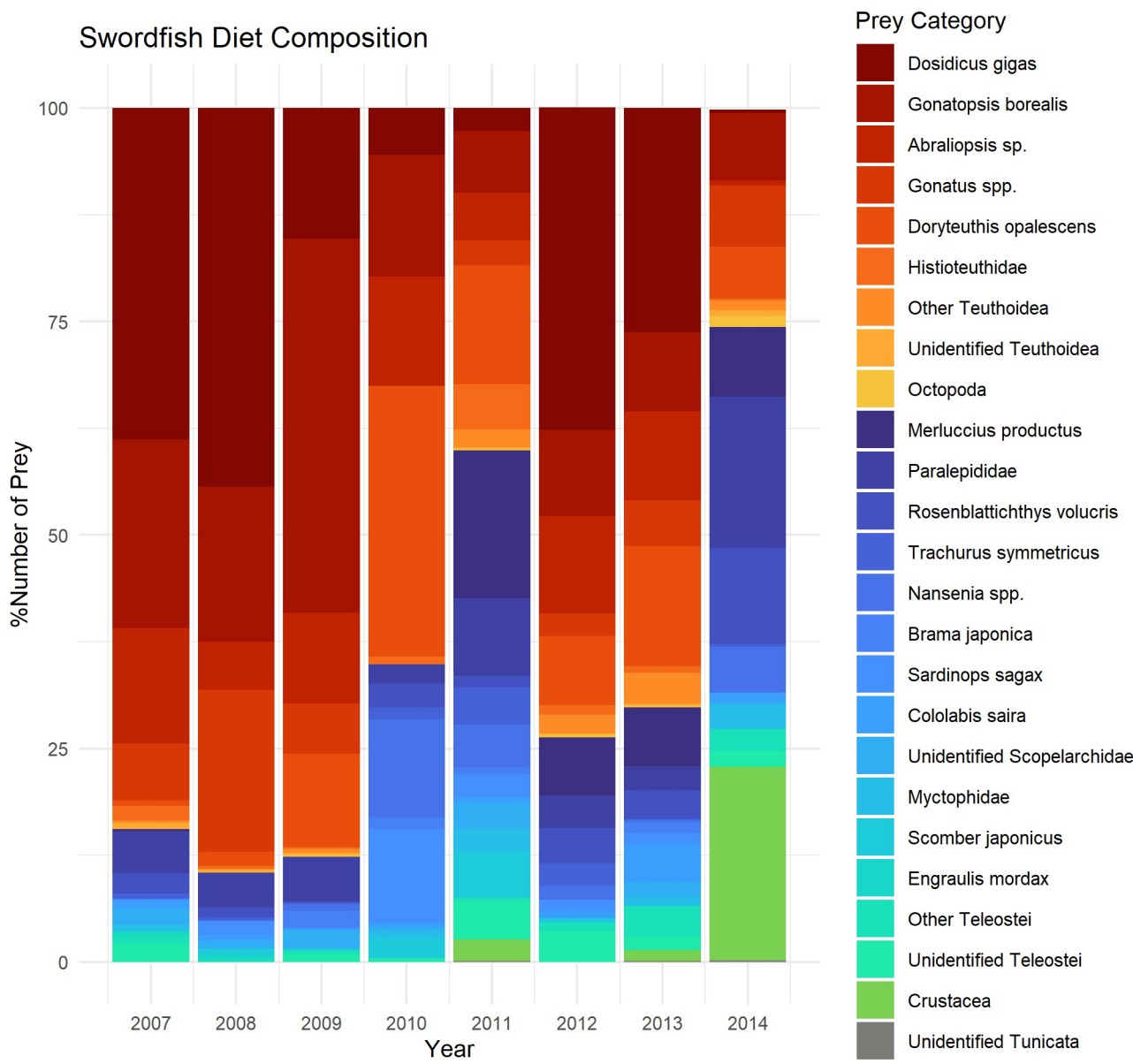

**Fig 4. Composition (%N) by year for swordfish diet components.** Red = Teuthoidea; Blue = Teleostei; Green = Crustacea; Grey = Tunicata.

components within each year from 2007–2014 are shown in Fig 4. Prey taxa were combined to limit their number for graphic purposes. Groupings by family, infraclass, or order were applied in some cases.

### Redundancy analysis (RDA)

Explanatory variables related to fish length (EFL), area, year and half-year, all significantly affected the overall pattern of variation in diet (numerical importance of prey) in swordfish. SST did not significantly affect any variation in diet (Table 4). Diet was significantly different (versus other years) in 2007 and 2011–2014. The set of explanatory variables used explained 6% of the overall variation in prey counts, with RDA axes 1 and 2 accounting for 36.9% and

**Table 4. Results of redundancy analysis (RDA) of variation in diet composition of swordfish (based on prey numbers).** Values of *F* and associated probability (*p*-value) are tabulated for two sets of model runs. The variable 'year' (fishing season) was divided into three categories (2007, 2008–2010 and 2011–2014) and converted into three (0,1) dummy variables. Since the category may be identified once the values of two of the dummy variables have been defined, all three dummy variables cannot be included in the same run of the model. Left: model runs excluding 2011–2014. Right: model runs excluding 2007. (EFL = eye to fork length, Area = within the SCB and beyond the SCB, Half-year = August 15th through November 7th and November 8th through January 31st).

| Variable | *F*-statistics | *p*-value | *F*-statistics | *p*-value |
|---|---|---|---|---|
| EFL | 4.117 | **0.005** | 4.254 | **0.005** |
| Area | 3.896 | **0.005** | 3.895 | **0.005** |
| 2007 | 3.383 | **0.005** | | |
| Half-year | 2.025 | **0.005** | 2.123 | **0.005** |
| 2011–2014 | | | 5.016 | **0.005** |
| 2008–2010 | 3.568 | **0.005** | 1.042 | 0.415 |
| SST | 0.758 | 0.785 | 0.758 | 0.815 |

23.1% of this variation respectively. The first two RDA axes thus explain around 3.8% of variation in prey counts, i.e., although significant temporal, spatial and size-related variation in diet has been demonstrated, the majority of observed dietary variation remains unexplained.

## Generalized Additive Models (GAMs)

To investigate sources of variation in the importance of individual prey taxa, negative binomial GAMs were fitted to count data for number of prey items in each stomach for the seven most important prey taxa, as ranked by GII. For jumbo squid, the final model contained significant effects of SST, EFL and year (Table 5). The presence of jumbo squid in swordfish stomachs was highest with SST around 21.5°C, it showed a linear increase with increasing swordfish length, and it was lowest in 2009 and highest in 2007 (Fig 5). The final model for *G. borealis* contained effects of year and area (Table 5). The presence of *G. borealis* in swordfish stomachs

**Table 5. Effect of explanatory variables on the presence of the main prey taxa in swordfish diet (form and direction of the relationship and statistical significance).** The first row for each species-variable combination contains the estimated degrees of freedom (edf) in the case of smoothers. The second row indicates the probability. Only significant effects, retained in the final models, are shown. Swordfish body length was measured as eye-to-fork length (EFL, cm). DE = deviance explained, AIC = value of the Akaike Information Criterion, R-sq (adj) = value of adjusted R-squared. Blank cells indicate non-significant effects that were dropped during model selection. 1st = first half of year, 2nd = second half of year; IN = within the SCB, OFF = beyond the SCB subregion.

| Swordfish | EFL | Year | SST | Half-year | Area | DE | AIC | R-sq (adj) |
|---|---|---|---|---|---|---|---|---|
| Jumbo squid | 1.0 (+) | 2.9 (∪) | 2.5 (+) | | | 25.0 | 1073.6 | 0.0561 |
| | P<0.0001 | P<0.0001 | P<0.0001 | | | | | |
| *Gonatopsis borealis* | | 2.9 (∩) | | | OFF>IN | 14.5 | 963.97 | 0.112 |
| | | P<0.0001 | | | P = 0.0105 | | | |
| *Abraliopsis* sp. | 1.0 (+) | 2.9 (∩) | | | | 9.8 | 727.51 | -0.00081 |
| | P = 0.0468 | P = 0.0031 | | | | | | |
| *Gonatus* spp. | | 2.8 (∪) | | 1st>2nd | | 13.4 | 632.83 | 0.0696 |
| | | P = 0.0058 | | P = 0.0049 | | | | |
| Market squid | | 2.8 (∩) | | | IN>OFF | 21.6 | 683.98 | 0.0589 |
| | | P<0.0001 | | | P = 0.0050 | | | |
| Pacific hake | 2.7 (+) | 2.0 (+) | | | | 26.6 | 355.48 | 0.0361 |
| | P = 0.0183 | P = 0.0004 | | | | | | |
| Duckbill barracudina | | 2.9 (∩) | | 2nd>1st | OFF>IN | 20.7 | 496.50 | 0.137 |
| | | P = 0.0002 | | P = 0.0097 | P = 0.0053 | | | |

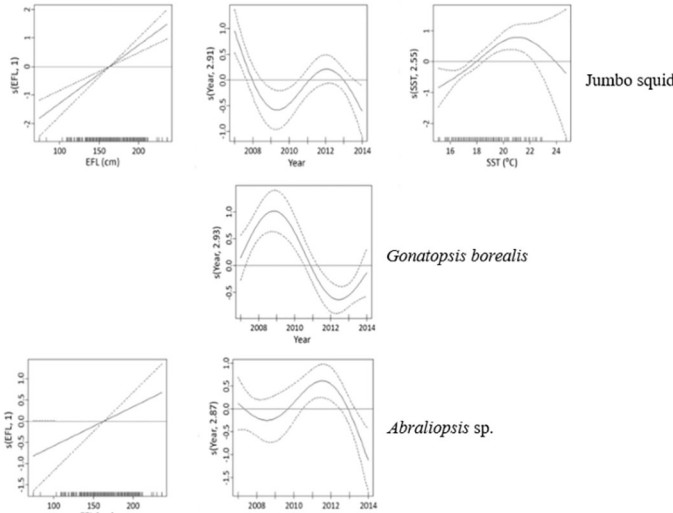

**Fig 5. GAM smoothing curves fitted to partial effects of explanatory variables on the presence of 3 prey taxa (jumbo squid, *Gonatopsis borealis*, *Abraliopsis* sp.) in the stomach of swordfish.** EFL = eye-to-fork length. Dashed lines represent 95% confidence intervals around the main effects.

was highest in 2009 and lowest around 2012 (Fig 5), and was higher beyond the SCB area than within.

For *Abraliopsis* sp., the final model contained effects of year and length (Fig 5). The presence of *Abraliopsis* sp. in swordfish stomachs was lowest in 2014 and highest in 2012, and showed a linear increase with increasing swordfish length (Fig 5). However, as indicated by the negative factor of adjusted R-squared, the model was unsatisfactory. For *Gonatus* spp. the final model contained effects of year and half-year (Table 5). The presence of *Gonatus* spp. in swordfish stomachs was highest around 2008–2009 and 2014 and was lowest in 2012 (Fig 6). Numbers of *Gonatus* spp. were higher in the first half-year (August 15 through November 7) than in the second (Table 5).

For market squid, the final model contained effects of year and area (Table 5). The presence of market squid in swordfish stomachs was highest in 2010 (Fig 6) and was higher within the SCB area than beyond it. For Pacific hake, the final model contained effects of year and length (Table 5). The presence of Pacific hake in swordfish stomachs was highest in 2012 and showed a positive relationship with fish length at lengths between around 125 and 150 cm (Fig 6). For duckbill barracudina, the final model contained effects of year, area, and half-year (Table 5). The presence of duckbill barracudina in swordfish stomachs was highest in 2009 (Fig 6). It was greater beyond the SCB area and during the second half of the fishing season (November 8 through January 31).

Residual plots for the seven most important prey taxa, as ranked by GII with respect to explanatory variables used in the selected GAMs models, are provided in S1–S3 Figs. Because the data represent small counts of individual species found in each stomach, residuals were not assumed to be normally distributed. By putting an implicit capacity limit on the number of prey items, stomach-level observations limit the potential presence of heteroscedasticity. The negative binomial model, a generalization of the Poisson distribution that does not assume the variance equals the mean, as appropriate for the count data, was used. Explanatory variables were included to account for known dependencies. The residual plots visually confirm the positive skewness / non-normality of the data and do not suggest the presence of heteroscedasticity.

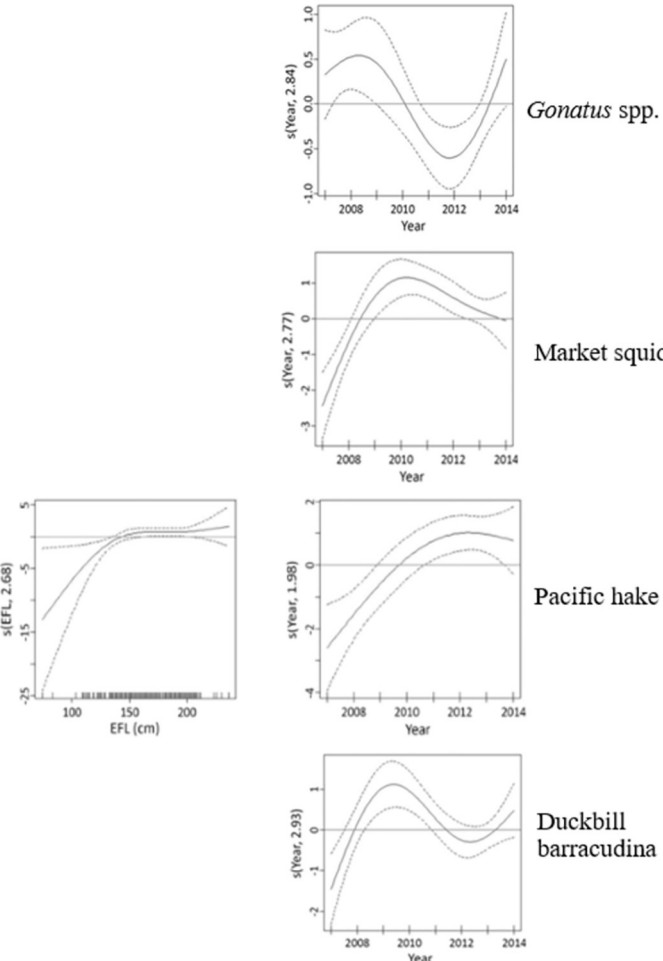

**Fig 6. GAM smoothing curves fitted to partial effects of explanatory variables on the presence of 4 prey taxa (*Gonatus* spp., market squid, Pacific hake, duckbill barracudina) in the stomach of swordfish.** EFL = eye-to-fork length. Dashed lines represent 95% confidence intervals around the main effects.

## Discussion

The range of prey species found in our study is consistent with the diurnal vertical distribution of swordfish, reflecting their diving behavior. Vertical movements allow pelagic predators to extend their prey base or access different resources. In marine ecosystems, diel changes in distribution or behavior of predators are frequently in tune with diel changes in prey distribution, such as vertical migration of organisms associated with the deep scattering layer (DSL) [77]. The diurnal vertical distribution of swordfish is region-specific and likely influenced by both abiotic (temperature, thermocline depth, dissolved oxygen) and biotic factors (prey abundance and distribution, body temperature) [20]. Swordfish can feed at great depths during diurnal vertical migrations [25] and can feed during both day and night within the DSL [78]. Electronic tagging studies on swordfish in the CCLME show that these predators are capable of exhibiting highly variable movements during the day but are consistently found within the upper mixed layer at night [20, 22]. These movements are consistent with those of the DSL.

Results of the present study indicate that swordfish fed mainly on cephalopods and teleosts, the most important prey taxa being jumbo squid (*Dosidicus gigas*), *Gonatopsis borealis* and

**Table 6. Proportion of teleosts and cephalopods, by area, in diet of swordfish based on published studies.** '*' = highest proportion; W = Western, N = North, E = Eastern, S = Southern, Teleo = teleosts, Ceph = cephalopods.

| Area | Teleo | Ceph | Authors |
|---|---|---|---|
| W. N. Atlantic | * | | [29, 87, 88, 106–118] |
| | | * | [80, 89] |
| E. N. Atlantic | * | * | [90, 119, 120] |
| | | * | [81, 91] |
| E. Central Atlantic | * | * | [92, 121] |
| E. Tropical Atlantic | | * | [93] |
| Tropical Atlantic | * | | [122] |
| W. Equatorial Indian Ocean | * | | [123] |
| E. N. Pacific (Channel Islands, California) | * | | [31] |
| E. N. Pacific (Baja California) | * | * | [29] |
| | | * | [30] |
| Central N. Pacific (Hawaii) | | * | [94] |
| E. Pacific (Chile) | | * | [124–128] |
| | * | | [129] |
| E. Pacific (Ecuador) | | * | [130–132] |
| S. Pacific | * | | [133] |
| W. N. Pacific | | * | [134] |
| W. Mediterranean Sea | * | | [135, 136] |
| E. Mediterranean Sea | | * | [82] |
| S. Aegean Sea | * | | [137] |
| E. Australia | | * | [83] |
| | * | | [138] |

*Abraliopsis* sp., while teleosts included both epipelagic and mesopelagic species. Results are thus in broad agreement with those from several studies of this species in other regions [29, 30, 79–83], although the relative importance of fish and cephalopods varies between different areas (see Table 6).

Jumbo squid was an important prey item for swordfish in the CCLME, as was also the case for several shark species (for mako, blue and bigeye thresher) in the area [5]. This finding is likely linked to the range expansion of jumbo squid that started around 2002 in the CCLME. These cephalopods, rarely found in the CCLME previously, greatly extended their range in the eastern North Pacific Ocean during a period characterized by ocean-scale warming, regional cooling, and the decline of tuna and billfish populations throughout the Pacific [84, 85]. Jumbo squid belongs to the Ommastrephidae, a family of largely pelagic squids that includes several species that support important commercial squid fisheries around the world [86]. Ommastrephids, in general, have been described as the most important cephalopod prey for swordfish in other regions of the world [28, 29, 80, 82, 87–94] in both coastal and pelagic ecosystems.

Of the squids eaten by swordfish, while gonatids and onychoteuthids, are mainly epipelagic and all are powerful swimmers, ommastrephids like jumbo squid and the histioteuthids are predominantly mesopelagic drifters [30, 95], indicating that swordfish can feed in different environments. Since swordfish detect their prey visually [25], swordfish may more easily catch fast-swimming, medium to large cephalopods than small, slow-moving prey [30]. Prey items with size measurements available ranged from 90 mm to 650 mm. The most frequent prey items presented an average length between 199 mm and 356 mm. Market squid was the smallest among the prey measured with an average size of 105 mm (Table 2).

Pacific hake was, overall, the most important teleost species in the diet, based on ranking by GII, followed by duckbill barracudina. Scombrids were also present in the diet. Merlucciids, paralepidids, and scombrids have been described as important fish prey species of swordfish in a number of other studies in different areas [28, 29, 31, 80, 87, 88, 90, 92]. All are abundant species in coastal pelagic ecosystems where swordfish are usually caught. Seven species of Myctophidae, two species of Scopelarchidae and one species of Bathylagidae were present in this study, indicating that swordfish forage frequently in mesopelagic waters.

A number of the most important swordfish prey species are found in or associated with the DSL, including jumbo squid, *G. borealis* and *Gonatus* spp. squids, barracudinas, and Pacific hake [96–101]. Other important prey, like *Abraliopsis* sp. and market squid, are more epipelagic. The range of prey species eaten, in terms of both prey size and prey habitat, suggests that swordfish have quite flexible foraging strategies.

The combination of large size, endothermy, and the lack of slicing teeth possibly places swordfish closer to dolphins rather than sharks in terms of foraging ecology. Swordfish diets and prey composition have been found to vary by ecosystem. In some regions, swordfish diets presented a prevalence of teleosts, while in others cephalopods were most prominent. In a few areas, a similar proportion of both prey item groups were observed (Table 6). Several studies considered only the cephalopod portion of the swordfish diet and, therefore, are not listed in Table 6 [79, 102–105].

GII and IRI are useful indices to get an overview of the importance of prey species. However, each of the three RMPQs used in the calculation of these indices has a different meaning. Frequency (F) reflects foraging opportunities. In the case of a predator that picks up individual prey items, such as swordfish, number (N) would reflect an aspect of prey availability and foraging effort; weight (W) would relate to its importance as an energy source. Therefore, in general, when estimating the importance of a prey item, it is necessary to analyze the RMPQs of each major prey species separately. For example, *Abraliopsis* sp., a small squid, is not an important prey in terms of W, although its high F and N indicate that it is fed on frequently. Nevertheless, it is important to note that GII and IRI are high. As an opposite pattern, luvar has a large body size, and even though the weight index is large, both F and N are low, suggesting that its importance as food is limited to a few individuals.

Results on importance of prey are based on GII and IRI calculated with 91% of prey that were in an advanced state of digestion. If prey had been in a more recent state of digestion, results could have been different. It is understood that using weight of prey remains can result in biased index calculations. Some diet studies use a reconstruction method where the relative prey importance and dietary composition are estimated from a back-calculation of the weight of every prey item based on identification and size measurements of body remains in the stomach [139–141]. Amundsen PA and Sánchez-Hernández J (2019) dispute this method as it tends to overestimate the role of prey that digest slowly [142]. This bias becomes larger when back-calculations are based on fish otoliths or squid beaks as these hard parts can stay in the stomachs for a long time and will lead to an overestimation of prey importance [143, 144].

## Dietary variation in swordfish

The importance of several prey taxa varied in relation to swordfish body size, location, year and, in some cases, differed between the first and second half of the fishing season. Jumbo squid, *Gonatus* spp. and Pacific hake were all more important as prey for larger swordfish than for smaller ones. At least in part, this may reflect the ability of larger swordfish to catch and eat large prey. These results differ from those of [29] who did not find variability in diet by size in swordfish off western Baja California.

Jumbo squid, *Gonatus* spp. and market squid were more important inshore (within the SCB) while *G. borealis*, Pacific hake and duckbill barracudina were more important offshore (beyond the SCB). These differences probably reflect prey availability but more information is needed on distribution of cephalopods and fish to confirm this.

Significant between-year variation in diet was also apparent. In general, this may reflect long-term variation in swordfish preference, prey availability, prey distribution, and prey abundance, but could also be related to changes in fishing locations. According to GII results, jumbo squid was more important in swordfish diet from 2007–2010 than in 2011–2014, with Pacific hake being the most important prey item in the latter period. However, GAM analysis shows a peak in jumbo squid for 2012, suggesting this species increased in dietary importance after 2010, once other factors are taken into account. These results likely relate to the range expansion of jumbo squid that occurred during the first decade of the 2000s and the subsequent decline to lower levels in 2010 in the CCLME [145]. A prolonged decline of jumbo squid landings was observed also in the Gulf of California after El Niño (2009–2010) and was associated with chronic low-wind stress and decreased chlorophyll a [146].

The presence of jumbo squid in swordfish stomachs indicated a positive influence of SST and was highest around 21.5˚C. Jumbo squid abundance and availability in the CCLME was strongly seasonal. Smaller animals have been observed to move up from Mexican waters in mid-late spring, further offshore, reaching the Pacific Northwest (and at times up to Alaska) in the summer, then slowly returning back down the coast in fall and early winter, when much larger, often closer to shore (but also in deeper waters) as they moved back to Mexico [147, 148]. In the northern hemisphere, jumbo squid are known to spawn in Mexican waters in the Gulf of California [149, 150] and off the Pacific coast of Baja California Sur [151, 152]. Other spawning grounds may exist; temperatures between 15–25˚C have been identified as permissive for proper development of paralarvae in the laboratory and are available seasonally offshore of California [153]. Following this scenario, the SST trend detected in the GAM might be a reflection of seasonal availability of jumbo squid in this strongly seasonal upwelling ecosystem. Other marine ecosystems such as tropical ones may exhibit different temperature relations.

*G. borealis*, *Gonatus* spp. and market squid were most important from 2008–2010, a period which included both (cold) La Niña conditions in 2008 and a (warm) El Niño event in 2010. The increased incidence of market squid in swordfish diet coincided with a high abundance of market squid in both midwater trawl surveys and in landings [154]. The commercial squid fishery in California targets spawning aggregations 1–3 km from the shore, around the Channel Islands and near coastal canyons. Catches are highly influenced by El Niño events [155, 156]. The cooler water during the La Niña years may have favored higher abundance and therefore higher catches in market squid [157]. *Gonatus* spp. was more important in the diet during the 1st half-year period while duckbill barracudina was more present during the 2nd half-year period. This could be due to seasonal variation in the presence of these prey species or in the spatial distribution fisheries effort.

Northern anchovy is a monitored species under the Pacific Fishery Management Council's Coastal Pelagic Species fishery management plan. It was only found in three stomachs in this study, inside the SCB in 2007 and 2008. Mearns et al (1981) examined the stomach contents of 15 swordfish caught near the Southern California Channel Island in fall/winter of 1980 and found that northern anchovy accounted for over 40% of IRI. These differences may be attributed to variations in anchovy abundance over the years. Anchovy were present in higher numbers in the California Current prior to 1990 with a peak in catches around 1980 [158]. Catch estimates show that, starting around 2009 to 2013, northern anchovy biomass dropped to low levels [159]. Analysis of northern anchovy stock size from 1951–2011 suggested that the population was near an all-time low from 2009–2011 [160], and subsequent analysis suggested that

the population remained low through 2015 [161]. More recent minimum abundance estimates based on acoustic trawl surveys indicate the combined biomass of the Northern and Central stocks rebounded to a range from 0.5 to 1.1 million metric tons in 2018 and 2019 [162, 163].

Pacific sardine (the abundance of which until recently was believed to vary inversely with that of anchovy) [164–166] was not present in the diet in 2007 and sardine %F was low for other years of the study. These results are possibly related to the low sardine biomass during the study period [167], but they could be explained also by limited swordfish preference for sardine. Markaida and Sosa-Nishizaki (1998) reported a low %F for sardine in the diet of swordfish from northern Baja California in 1992–1993, a period when sardine biomass was higher in the area.

Future diet studies on swordfish in the CCLME would benefit from more information on prey distribution and abundance (and thus their availability to swordfish) and on the size distribution of available and consumed prey. This would potentially allow elucidation of (multivariate) functional responses (i.e., how numbers of a prey species in the diet relate to its abundance and the abundance of other prey species) [168].

The present study would have benefited from a larger sample size since the rarefaction curve for number of prey species detected versus sample size did not reach an asymptote. In Bizzarro et al (2007), prey taxa were grouped into a limited number of categories causing several curves in their study to reach true asymptotes. Identifying most of the prey items in this study to the species level made reaching an asymptote more difficult than if the Bizzarro et al (2007) approach had been followed, due to a potentially large number of ungrouped individual species with small counts. The curve in this study approaches the asymptote, indicating that the most important prey items were included. More stomach samples would be required to cover the entire spectrum of less frequently encountered prey items, and authors are in the process of collecting additional data.

Samples used in this study were collected during the fall/winter period and were fisheries-dependent so information on the diet at other times of the year is lacking. Results are also potentially influenced by the distribution and targeting of fisheries effort and catch. While additional studies are warranted, this study provides the most comprehensive view of swordfish diets in the CCLME to date, allowing for comparisons of diet in relation to size, year and area.

## Supporting information

**S1 Fig. Residual plots for (jumbo squid, *Gonatopsis borealis*, *Abraliopsis* sp.) with respect to explanatory variables used in the selected GAMs represented in Fig 5.**
(TIF)

**S2 Fig. Residual plots for (*Gonatus* spp., market squid, Pacific hake) with respect to explanatory variables used in the corresponding selected GAMs represented in Fig 6.**
(TIF)

**S3 Fig. Residual plots for (duckbill barracudina) with respect to explanatory variables used in the corresponding selected GAMs represented in Fig 6.**
(TIF)

**S1 Table. Quantitative prey composition of the broadbill swordfish (EFL < 165 cm) in the California current.** A total of 148 stomachs containing food was examined. Prey items are shown by decreasing GII value. See methods for description of the measured values.
(DOCX)

**S2 Table. Quantitative prey composition of the broadbill swordfish (EFL ≥ 165 cm) in the California current.** A total of 140 stomachs containing food was examined. Prey items are shown by decreasing GII value. See methods for description of the measured values.
(DOCX)

**S3 Table. Comparison of GII for the main prey species between small and medium broadbill swordfish.** Values of mean GII, bootstrapped 95% CIs and % bootstrap runs in which each prey type was in the smaller of two size categories of swordfish. If more than 95% (or fewer than 5%) of runs show the prey type was more important in the smaller size category of swordfish than in the larger category, the difference is considered to be significant. S = small (EFL < 165 cm), M = medium (EFL ≥ 165 cm). These results are generally consistent with inferences from non-overlap of 95% CIs.
(DOCX)

**S4 Table. Quantitative prey composition of the broadbill swordfish within the SCB subregion.** A total of 199 stomachs containing food was examined. Prey items are shown by decreasing GII value. See methods for description of the measured values.
(DOCX)

**S5 Table. Quantitative prey composition of the broadbill swordfish beyond the SCB subregion.** A total of 93 stomachs containing food was examined. Prey items are shown by decreasing GII value. See methods for description of the measured values.
(DOCX)

**S6 Table. Comparison of GII for the main prey species between broadbill swordfish within and beyond the SCB region.** Values of mean GII, bootstrapped 95% CIs and % bootstrap runs in which each prey type was in each of two categories of swordfish. If more than 95% (or fewer than 5%) of runs show the prey type was more important in one region than the other, the difference is considered to be significant. East = within the SCB subregion, West = beyond the SCB subregion. These results are generally consistent with inferences from non-overlap of 95% CIs.
(DOCX)

**S7 Table. Quantitative prey composition of the broadbill swordfish during year 2007 in the California current.** A total of 47 stomachs containing food was examined. Prey items are shown by decreasing GII value. See methods for description of the measured values.
(DOCX)

**S8 Table. Quantitative prey composition of the broadbill swordfish during year 2008 in the California current.** A total of 16 stomachs containing food was examined. Prey items are shown by decreasing GII value. See methods for description of the measured values.
(DOCX)

**S9 Table. Quantitative prey composition of the broadbill swordfish during year 2009 in the California current.** A total of 37 stomachs containing food was examined. Prey items are shown by decreasing GII value. See methods for description of the measured values.
(DOCX)

**S10 Table. Quantitative prey composition of the broadbill swordfish during year 2010 in the California current.** A total of 12 stomachs containing food was examined. Prey items are shown by decreasing GII value. See methods for description of the measured values.
(DOCX)

**S11 Table. Quantitative prey composition of the broadbill swordfish during year 2011 in the California current.** A total of 54 stomachs containing food was examined. Prey items are shown by decreasing GII value. See methods for description of the measured values. (DOCX)

**S12 Table. Quantitative prey composition of the broadbill swordfish during year 2012 in the California current.** A total of 36 stomachs containing food was examined. Prey items are shown by decreasing GII value. See methods for description of the measured values. (DOCX)

**S13 Table. Quantitative prey composition of the broadbill swordfish during year 2013 in the California current.** A total of 56 stomachs containing food was examined. Prey items are shown by decreasing GII value. See methods for description of the measured values. (DOCX)

**S14 Table. Quantitative prey composition of the broadbill swordfish during year 2014 in the California current.** A total of 34 stomachs containing food was examined. Prey items are shown by decreasing GII value. See methods for description of the measured values. (DOCX)

**S15 Table. Comparison of GII for the main prey species for broadbill swordfish by year group.** Values of mean GII, bootstrapped 95% CIs and % bootstrap runs in which each prey type was in each of two categories of swordfish. If more than 95% (or fewer than 5%) of runs show the prey type was more important in one year than the other, the difference is considered to be significant. Y1 = Year1 (2007), Y2 = Year2 (2008–2010), Y3 = Year3 (2011–2014). These results are generally consistent with inferences from non-overlap of 95% CIs. (DOCX)

## Acknowledgments

This work would not have been possible without the assistance and samples provided by the NMFS Southwest Region Fishery Observer Program and the participating drift gillnet fishermen. We thank several assistant volunteers who helped process the stomach samples. Alexandra Stohs provided research assistance. Mark Lowry, Eric Hochberg and John Hyde helped identify some prey specimens. John Field, Chugey Sepulveda and Scott Aalbers offered science feedback. Barbara Muhling helped create the map. Kristen Koch, Annie Yau, Brad Erisman, Heidi Dewar, Stephanie Flores, Crystal Dombrow, Elan Portner and Ruben Bergtraun provided useful comments on the draft. Debra Losey assisted with library research. We also thank Hiroshi Ohizumi and two anonymous reviewers for their careful critiques that helped improve the manuscript.

## Author Contributions

**Conceptualization:** Antonella Preti, Graham J. Pierce.

**Data curation:** Antonella Preti, Stephen M. Stohs, Graham J. Pierce.

**Formal analysis:** Antonella Preti, Stephen M. Stohs, Camilo Saavedra, Graham J. Pierce.

**Funding acquisition:** Gerard T. DiNardo.

**Investigation:** Antonella Preti.

**Methodology:** Antonella Preti, Stephen M. Stohs, Camilo Saavedra, Graham J. Pierce.

**Project administration:** Antonella Preti.

**Resources:** Antonella Preti.

**Software:** Antonella Preti, Stephen M. Stohs, Camilo Saavedra, Graham J. Pierce.

**Supervision:** Stephen M. Stohs, Gerard T. DiNardo, Ken MacKenzie, Leslie R. Noble, Catherine S. Jones, Graham J. Pierce.

**Validation:** Antonella Preti.

**Visualization:** Antonella Preti.

**Writing – original draft:** Antonella Preti.

**Writing – review & editing:** Antonella Preti, Stephen M. Stohs, Gerard T. DiNardo, Camilo Saavedra, Ken MacKenzie, Leslie R. Noble, Catherine S. Jones, Graham J. Pierce.

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
