## [Decision Letter · Decision Letter 0]

9 Nov 2021

PONE-D-21-29802Feeding ecology of broadbill swordfish (Xiphias gladius) in the California Current

PLOS ONE

Dear Dr. Preti,

Thank you for submitting your manuscript to PLOS ONE. After careful consideration, we feel that it has merit but does not fully meet PLOS ONE’s publication criteria as it currently stands. Therefore, we invite you to submit a revised version of the manuscript that addresses the points raised during the review process.

Three expert reviewers feel that your manuscript is a valuable contribution, worthy of publication. However, they have raised some concerns with a number of aspects that must be addressed and resolved before the manuscript can be considered for publication. In particular, two reviewers believe that the novelty of your contribution is not well reflected in the manuscript. The introduction should be shortened as there is unnecessary (non-relevant) information (e. g., fishery descriptions). There are also several methodological improvements to be made (results of DNA prey identification, GAM analysis descriptions, sample size vs statistical robustness, etc.). A few relevant references related to the research topic have not been cited/discussed.

A marked-up copy of your manuscript that highlights changes made to the original version. You should upload this as a separate file labeled 'Revised Manuscript with Track Changes'.An unmarked version of your revised paper without tracked changes. You should upload this as a separate file labeled 'Manuscript'.

We look forward to receiving your revised manuscript.

Kind regards,

Antonio Medina Guerrero, Ph.D.

Academic Editor

PLOS ONE

Journal Requirements:

2. In your Methods section, please provide additional information regarding the permits you obtained for the work. Please ensure you have included the full name of the authority that approved the field site access and, if no permits were required, a brief statement explaining why

3. We note that Figure 1 in your submission contain map images which may be copyrighted. All PLOS content is published under the Creative Commons Attribution License (CC BY 4.0), which means that the manuscript, images, and Supporting Information files will be freely available online, and any third party is permitted to access, download, copy, distribute, and use these materials in any way, even commercially, with proper attribution. For these reasons, we cannot publish previously copyrighted maps or satellite images created using proprietary data, such as Google software (Google Maps, Street View, and Earth). For more information, see our copyright guidelines: http://journals.plos.org/plosone/s/licenses-and-copyright.

Reviewers' comments:

Reviewer's Responses to Questions

**Comments to the Author**

1. Is the manuscript technically sound, and do the data support the conclusions?

Reviewer #1: Partly

Reviewer #2: Yes

Reviewer #3: Yes

2. Has the statistical analysis been performed appropriately and rigorously? 

Reviewer #1: No

Reviewer #2: Yes

Reviewer #3: Yes

3. Have the authors made all data underlying the findings in their manuscript fully available?

Reviewer #1: Yes

Reviewer #2: Yes

Reviewer #3: Yes

4. Is the manuscript presented in an intelligible fashion and written in standard English?

Reviewer #1: Yes

Reviewer #2: Yes

Reviewer #3: Yes

5. Review Comments to the Author

Reviewer #1: General comment:

The manuscript present the description of the dietary habits of Swordfish in the California Current, using stomach content analysis of individuals collected during several years. The introduction is well written with a good state of the art and objectives, but few sentences should be improved or amended (see comments below). Since previous studies already conducted stomach content analysis of Swordfish, the novelty and relevance of the study should be better stated in the introduction and discussion. The information presented is valuable, but before considering the manuscript for publication some of the analysis should be reconsidered or better described. My main concerns are, the division of small and large individuals, without a biological criteria, the lack of results regarding the DNA identification of prey and the diagnosis of the GAM analysis and the explanation of how variables were included in the models. The discussion section is very well written and easy to follow. Overall, I think the paper is interesting and has the potential to be published, but before, major revisions are needed. Please, find below some changes/comments/suggestions to consider in revising their manuscript. Since, I am not an English native speaker I have not evaluated the English.

Abstract

Line 37: I am not a native English speaker and I might be wrong, but in the sentence “with Pacific hake the most important prey item in the latter period” I think a verb is missing and should be something like “with Pacific hake being the most important prey item in the latter period”

Introduction

Line 47-50: To avoid repeating the concept and the mention to shark and dolphins merge the two sentences.

Line 64: Some literature regarding swordfish migration patterns is available in peer-reviewed publications. For example: Sepulveda et al., (2020) Insights into the horizontal movements, migration patterns, and stock affiliation of California swordfish. Fisheries Oceanography, DOI: 10.1111/fog.12461. Please, modify the sentence accordingly.

Line 91: Sharks include many species with different foraging ecology and trophic level. Perhaps the sentence of lines 90-91 can be removed or better explained including references.

Table 1: Since the proportions are not reported in the table, please change the Table description for a more accurate text. For example: “Prey group dominance” or include the proportion values reported in the literature. Also, if you want to include a information on the diet of the western Mediterranean you have this one: Navarro et al., (2017) Feeding strategies and ecological roles of three predatory pelagic fish in the western Mediterranean Sea. https://doi.org/10.1016/j.dsr2.2016.06.009

Line 107-108: Please, highlight the novelty and the need of the present manuscript in comparison with the previous studies cited.

Line 111-112: Be more concrete, how the findings can serve to alternative approaches of management?

Material and Methods

Lines 151-153: If preys classified in digestive state 1 or 2 were discarded in the analysis, I assume that also they were not included in the weight of the undigested items mentioned in lines 145 or 146. If so, please mention that part (lines 151-153) before the line 144.

Lines 154-164: The use of genetic analysis to identify diet items is an interesting approach and since you have done this effort to better identify the stomach content, I recommend to highlight that you use DNA analysis in the abstract and even it could be mentioned in the end of the Introduction.

Line 215-219: Ontogenetic changes in diet tend to be mainly observed between juveniles and adults due to changes in energy requirements (growth vs. reproduction). In previous research it has been reported that first size of maturity is quite different between males and females. Did you have information related to sex to check if changes in diet with size are different between males and females? Because to classify small and large at 165 without any biological criteria might be not the best approach. I understand that it was done to have a balanced groups, but even thought might be not correct.

Line 224-227: It is not clear if the statistical analysis were only used the 6 most important prey items or it was done for all the diet prey. Many times differences in diet between groups of the same species are driven by not the main prey, so I recommend to include as many prey as possible in the statistical analysis.

Line 224-225: After reading the results I realized that each of the factors Size, Area, Year is analyzed separately/independently, one by one. Please, specify it in the text.

Line 237: Only 5 of the seven explanatory variables are mentioned (Area, year, half-year, predator size, SST). Which are the two remaining ones? Also justify why you the half year variable and which is the criteria to separate up to November 7. In line 218 -2019 you named the variables east and west ad Within SCB and Beyond SCB, be consistent. The same for the explanatory variable that in line 238 is called time-period, but then is named “Year”.

Line 240: If you group years the objective is to have a higher number of samples for each category, but you are not reducing the number of explanatory variables, since the variable “year” is still in the analysis. The text should be corrected.

Line 236-237: The effects on what?

Line 247: Which are the variables included in the GAM analysis? Which of the variables were included as continuous variables and which as factors? Did you include the half year nested within Year?

Did you performed a diagnosis of the model assumptions verifying the independence of the residuals for the theoretical assumption of normality, homoscedasticity and independence? Please, include the residual plots as supplementary material and a table with the model tested and the corresponding AIC and R2.

RESULTS

Figure 1: The Map should be modified including more information. In the map indicates the name of the pacific Ocean, the countries, or some geographical details that somebody who is not familiar with the area can easily interpreted. In the figure caption you mention “Baja California) but is not obvious where it is. Describe what it means the vertical line, and also describe in the figure, and in the figure caption the what it means the scale (Nº of individuals). In the figure caption include what it means CCLME, since each figure should be self-explicative without the need of reading the manuscript.

Lines 273-277: Connected with my previous comment, with figure 2 it is clearly stated that for females the group below 165cm includes mature and immature individuals. Then it is not justify to separate in two groups with the criteria of having the same number of samples.

Table 2: The information “ A total of 299 stomach was examined” is irrelevant here, mention that the table 2 correspond to the 292 individuals.

Comment: Can you provide a table with the number of samples (stomachs) included per year, area, and size group?

Lines 347-354: Describe also that SST was not significant

Line 365: The criteria of AIC and Deviance explained to select models should be explained in Material and methods. Also A supplementary table with the AIC of and DE of the background selection procedure to justify the model selection should be provided.

Line 365: Please, could you better explain in material and methods and results which data are you using in the GAM models as response variable? As it is right now is confusing. You run the models at individual stomach level? But then I don’t understand if it is presence absence, like stated in line 367, why it is said that was ranked by GII, but in Material and methods is indicated that was count data (number of preys items in each stomach).

Figure 4. If is a single figure with several panels, it should appear as a single figure. Now is split in 7 different figures. Also, in the figure caption it should be described to which species correspond each letter.

In the results I am missing the results from genetic identification. It would be interesting to include which were the species that thanks to including DNA analysis could be incorporated to understand the bias that is done in the description of diets when not including this type of identification.

Discussion:

From line 405 to 427 the text should be removed since is information not supported with your results. If some of the information provided in this two paragraphs you think is important to be mentioned, it can be moved to Introduction. The Discussion should start in line 428.

Line 459: The range of prey size in material an methods was indicated that it was measured, but then in results I couldn’t find anything reporting this results. It would be very interesting to have this information available.

The discussion in general is well written and interesting, but the results from GAM regarding the variables SST and Half-year are not discussed.

I know that the diet is already reported in tables, but a graph with the diet by year period would facilitated the intercomparison of changes in diet between years to better follow the discussion. But it is up to the authors to included or not, since I understand that this information is already provided.

Reviewer #2: This paper analyzes the feeding habits of swordfish using a large number of stomach content data. Careful statistical methods were used in the analysis of the data, and several environmental factors were successfully quantified to determine the extent to which their effects were reflected in the diet. I think this paper deserves to be published because I believe that such analysis and results will contribute greatly to our understanding of swordfish ecology.

However, I think there are still a few things that need to be added or revised in the current manuscript.

I think there is a lot of information, especially in the introduction, that is not directly necessary for discussing the diet of swordfish. However, this may be a prerequisite knowledge for the wide readership of PlosOne. Even so, I think the description of the fishery from page 2, line 51 to page 3, line 60 could probably be simplified. In addition, page 4, lines 91-95 and Table 1 should be moved to Discussion to consider the geographic variation in swordfish diet.

　I agree, that GII and IRI are useful to get an overview of the importance of prey species. However, I think each of the three RMPQs used in this calculation has a different meaning. For example, F would reflect foraging opportunities. In the case of a predator that picks up individual prey items, such as swordfish, N would reflect an aspect of prey availablity and foraging effort; W would relate to its importance as an energy source. Therefore, I think it is necessary to consider the RMPQs of each major prey species separately. For example, Abraliopsis, a small squid, is not an important prey in terms of weight, although its high F and N indicate that it is fed frequently. Nevertheless, it is important to note that GII and IRI are high. As an opposite pattern, luvar has a large body size, and even though the weight index is large, both F and N are low, suggesting that its importance as food is limited to a few individuals.

　In this manuscript, the first part of the discussion discusses the overall characteristics of the species that swordfish use as their primary food item based on GII. It concludes that the swordfish will have a flexible choice of medium or large prey items related to DSL. I think this is a reasonable conclusion. However, I think it would be desirable to be a little more careful in the discussions that lead to this conclusion. For example, with regard to the size of the prey species, it would be good to consider the characteristics of the main prey species in the list of prey species that were actually preyed upon, and come to a conclusion as to what size range the swordfish considers to be its main prey. In the current manuscript, this is only briefly described on page 23, lines 444 to 448. Similarly, I think the authors should distinguish between DSL-related mesopelagic prey and other epipelagic prey on the list before concluding on foraging depth. This careful treatment is especially necessary for PlosOne readers. As shown in Figure 3, I believe that present study has achieved a near upper limit of prey variation, although it is not perfect. Therefore, I think that such an evaluation of the diet overview is important in describing the general dietary characteristics of swordfish. Also, on page 4, line 85 of the preface, the authors describe the need for energy-rich food, but have you seen any general trend toward that?

Other miscellaneous points;

P8 lines140-143: What is the reference to identify fish otolith and squid beaks? Please specify literature or reference sample collection.

P10 lines 193-194: Empty stomachs were excluded from contribution analysis, but empty stomachs should be included in the calculation of frequency of occurrence.

P14 lines 278-279: 91% of the food items were severely digested. To what extent is this expected to affect the weight composition of the prey?　The extent to which it affects the assessment of importance by GII should be added to the discussion.

P19 Table 3: SST was not significant factor in the results of RDA, but in P20 lines 368-369, SST was included final model of GAM for jumbo squid. How can this inconsistency be interpreted?

Table S15: Y1 = S1?

Reviewer #3: COMMENTS TO AUTHORS

This manuscript addresses the food habits of swordfish (Xiphias gladius) in the California Current. The study is based on the analysis of gut contents and a complete set of data analyses derived from the stomach content analysis. Although the paper provides valuable information on the trophic biology of this species, I have found several deficiencies which should be amended. Some suggestions are given below.

Abstract

Lines 26-27. Remove “… federal….boats…”

Line 35. Remove “…in swordfish diet…”

Line 36. Change “form” to “in”

Line 37. Change “with Pacific hake the…” to “Pacific hake being”

Line 40. Change “from to “during”

Line 41. Which factors? Please clarify.

Line2 41-42. Standarditation? Please clarify.

Introduction

Lines 48-50. “Sowrdfish… dolphins” Please rewrite.

Line 50. Change “command” to “have”.

Lines 53-54. “Swordfish…fishery”. Please rewrite.

Line 64. Horizontal and vertical movements of the swordfish in the southern Pacific Ocean have been studied by Abascal et al (2010) and Evans et al (2014).

References:

- Abascal FJ, Mejuto J, Quintans M, Ramos-Cartelle A (2010) Horizontal and vertical movements of swordfish in the Southeast Pacific. ICES J Mar Sci 67:466–474

- Evans K, Abascal F, Kolody D, Sippel T, Holdsworth J, Maru P (2014) The horizontal and vertical dynamics of swordfish in the South Pacific Ocean. J Exp Mar Biol Ecol 450:55–67

Line 77. Swordfish can reach depths of up to 1136m (Abascal et al, 2010)

Lines “106-110”. According to the authors, there are 5 previous studies aimed to investigate the feeding habits of the swordfish in the area. Which is the novelty of the present study? In my opinion, the authors should provide information on the consumption rate using the model proposed by Olson and Mullen (1986). See also Olson and Boggs (1986) and Olson and Galván-Magaña (2002).

References:

- Olson RJ, Boggs CH (1986) Apex predation by yellowfin tuna (Thunnus albacares): independent estimates from gastric evacuation and stomach contents, bioenergetics, and cesium concentrations. Can J Fish Aquat Sci 43:1760–1775

- Olson RJ, Galván-Magaña F (2002) Food habits and consumption rates of common dolphinfish (Coryphaena hippurus) in the eastern Pacific Ocean. Fish Bull 100:279–298

- Olson RJ, Mullen AJ (1986) Recent developments for making gastric evacuation and daily ration determinations. Environ Biol Fish 16: 183–191

Table 1. The following studies have not included in this table.

- Abid N, laglaoui A, Arakrak A, Bakkali M (2018) The role of fish in the diet of swordfish (Xiphias gladius) in the Strait of Gibraltar. J Mar Biol Assoc UK 4: 895-907

- Holts D, Sosa-Nishizaki O (1998) Swordfish, Xiphias gladius, fisheries of the eastern North Pacific Ocean. In: Barret I, Sosa-Nishizaki O, Bartoo N (eds) Biology and fisheries of swordfish, Xiphias gladius. Papers from the International Symposium on Pacific Swordfish, Ensenada, Mexico, 11–14 December 1994. US Department of Commerce, NOAATechnical Report NMFS 142, pp 65–76.

- Logan JM, Golet W, Smith SC, Neilson J, Van Guelpen L (2021) Broadbill swordfish (Xiphias gladius) foraging and vertical movements in the north-west Atlantic. J Fish Biol 99: 557-568

- Young JW, Lansdell MJ, Campbell RA, Cooper SP, Juanes F, Guest MA (2010) Feeding ecology and niche segregation in oceanic top predators off eastern Australia. Mar Biol 157:2347–2368

- Zambrano-Zambrano RW, Mendoza-Moreira PE, Gómez-Zamora W, Varela JL (2019) Feeding ecology and consumption rate of broadbill swordfish (Xiphias gladius) in Ecuadorian waters. Mar Biodiver 49:373-380

Material and methods

Line 141. Which taxonomic keys?

Lines 145-150. In a recent review article, Amundsen and Sánchez-Hernández (2019) have criticized the fact of estimating prey weight from measurements of hard parts. Please, include it in Discussion section.

Reference

Amundsen P-A, Sánchez-Hernández J (2019) Feeding studies take guts – critical review and recommendations of methods for stomach contents analysis in fish. J Fish Biol 95:1364-1373

Line 197. According to Brown et al (2013), PSIRI provide more accurate estimates than IRI. Please, calculate PSIRI.

Reference:

Brown SC, Bizzarro JJ, Cailliet GM, Ebert DA (2012) Breaking with tradition: redefining measures for diet description with a case study of the Aleutian skate Bathyraja aleutica (Gilbert 1896). Environ Biol Fish 95:3-20

Provide information on how feeding consumption rate has been estimated. According to Olson and Galván-Magaña (2002) weight data estimated form hard parts (cephalopod beaks and/or fish otoliths) should not be considered in this analysis.

Reference:

- Olson RJ, Galván-Magaña F (2002) Food habits and consumption rates of common dolphinfish (Coryphaena hippurus) in the eastern Pacific Ocean. Fish Bull 100:279–298

Line 227. Why did you only use GII values for the analysis? Please explain.

Results

Provide consumption rate data by size class and area.

Lines 274. Change “[109]” to “authors et al [109]

Lines 274-277 “[109]… determined”. Remove it. As the authors have stated, they did not record fish sex.

Lines 291-296. Considering the p value reported by the authors, it looks like the number the samples were not enough to describe the diet completely. In fact, the authors stated in Discussion section that this study “would have benefited from a large sample size since” the curve did not reach the asymptote. Please, indicate how the low number the samples may affect the robustness of your results.

Lines 302-313. Please provide the values of the alimentary indices (when possible).

Lines 326-345. Why did you only compare GI values, but no IRI (or PSRI)? Clarify.

A table including number the samples collected by area, size class and year is missed. In this table, the authors should also include the number of samples with at least one prey by group (and %percentage of non-empty stomachs)

Discussion

Lines 414-417. “Adult…whales” Remove it. This information is not in line with the manuscript.

Line 444. According to Gilly et al (2006), the jumbo squid is a mesopelagic cephalopod. Please

check the following MS:

- Gilly W.F., Markaida U., Baxter C.H., Block B.A., Boustany A., Zeidberg L., Reisenbichler K., Robison B., Bazzino G. and Salinas C. (2006) Vertical and horizontal migrations by the jumbo squid Dosidicus gigas revealed by electronic tagging. Marine Ecology Progress Series 324, 1–17.

To enrich Discussion section, feeding consumption data should be compared to those reported in previous studies (Stillwell and Kohler, 1985: Young et al., 2010; Zambrano-Zambrano et al., 2019).

References

- Stillwell CE, Kohler NE (1985) Food and feeding ecology of the swordfish Xiphias Gladius in the western North Atlantic Ocean with estimates of daily ration. Mar Ecol Prog Ser 22:239–247

- Young JW, Lansdell MJ, Campbell RA, Cooper SP, Juanes F, Guest MA (2010) Feeding ecology and niche segregation in oceanic top predators off eastern Australia. Mar Biol 157:2347–2368

- Zambrano-Zambrano RW, Mendoza-Moreira PE, Gómez-Zamora W, Varela JL (2019) Feeding ecology and consumption rate of broadbill swordfish (Xiphias gladius) in Ecuadorian waters. Mar Biodiver 49:373-380

6. PLOS authors have the option to publish the peer review history of their article (what does this mean?). If published, this will include your full peer review and any attached files.

Reviewer #1: No

Reviewer #2: No

Reviewer #3: No

---

## [Author Response · Author response to Decision Letter 0]

3 Jun 2022

We added a file "Response to Reviewers" with detailed information.

We copy the file below.

Dr. Antonio Medina Guerrero

Academic Editor

PLOS ONE

Dear Dr. Medina Guerrero, 

Thank you for the opportunity to submit a revised draft of our manuscript titled: “Feeding ecology of broadbill swordfish (Xiphias gladius) in the California Current”. The preprint has generated considerable interest, and we are looking forward to formally publishing the article. We appreciate the time and effort that you and the reviewers have dedicated to providing your valuable feedback and insightful comments to our manuscript. We have incorporated changes to reflect most of the suggestions reviewers provided. We have highlighted our proposed changes to the manuscript through MS Word’s ‘track changes’ feature. Here is a point-by-point response to the reviewers’ comments and concerns.

We appreciate your further work to consider our proposed revisions.

Sincerely,

Antonella Preti and coauthors

We double checked our text and hope to have satisfied all style requirements.

2. In your Methods section, please provide additional information regarding the permits you obtained for the work. Please ensure you have included the full name of the authority that approved the field site access and, if no permits were required, a brief statement explaining why. Our samples are stomachs collected from fish captured in the U.S. west coast commercial swordfish fishery. They are considered fisheries discards which require no special permission to utilize in research. We added a sentence to the methods description to clarify this.

3. We note that Figure 1 in your submission contain map images which may be copyrighted. 

Our project team used the base R plot command to produce this figure. There is no copyright infringement.

Comments from Reviewer #1

Line 37: I am not a native English speaker and I might be wrong, but in the sentence “with Pacific hake the most important prey item in the latter period” I think a verb is missing and should be something like “with Pacific hake being the most important prey item in the latter period”. 

We added a verb as suggested. 

“Line 47-50: To avoid repeating the concept and the mention to shark and dolphins merge the two sentences.” 

We merged the two sentences as suggested to avoid repetition.

Line 64: Some literature regarding swordfish migration patterns is available in peer-reviewed publications. For example: Sepulveda et al., (2020) Insights into the horizontal movements, migration patterns, and stock affiliation of California swordfish. Fisheries Oceanography, DOI: 10.1111/fog.12461. Please, modify the sentence accordingly. 

We added a sentence on the migration patterns and included the requested reference.

Line 91: Sharks include many species with different foraging ecology and trophic level. Perhaps the sentence of lines 90-91 can be removed or better explained including references. 

We agree with the suggestion and have removed the sentence.

Table 1: Since the proportions are not reported in the table, please change the Table description for a more accurate text. For example: “Prey group dominance” or include the proportion values reported in the literature. Also, if you want to include a information on the diet of the western Mediterranean you have this one: Navarro et al., (2017) Feeding strategies and ecological roles of three predatory pelagic fish in the western Mediterranean Sea. https://doi.org/10.1016/j.dsr2.2016.06.009

We added “prey group dominance” to the caption. We further added a row in Table 1 (which is now retitled Table 6), and included the recommended reference.

Line 107-108: Please, highlight the novelty and the need of the present manuscript in comparison with the previous studies cited. 

We added some sentences to explain in detail what new features are presented in the study.

Line 111-112: Be more concrete, how the findings can serve to alternative approaches of management? 

We added a paragraph that explains in detail a number of potential applications of diet information in ecosystem-based management.

Material and Methods

Lines 151-153: If preys classified in digestive state 1 or 2 were discarded in the analysis, I assume that also they were not included in the weight of the undigested items mentioned in lines 145 or 146. If so, please mention that part (lines 151-153) before the line 144. 

We moved lines 151-153 to before line 144 as suggested.

Lines 154-164: The use of genetic analysis to identify diet items is an interesting approach and since you have done this effort to better identify the stomach content, I recommend to highlight that you use DNA analysis in the abstract and even it could be mentioned in the end of the Introduction. 

We have highlighted the DNA analysis in the abstract. 

Line 215-219: Ontogenetic changes in diet tend to be mainly observed between juveniles and adults due to changes in energy requirements (growth vs. reproduction). In previous research it has been reported that first size of maturity is quite different between males and females. Did you have information related to sex to check if changes in diet with size are different between males and females? Because to classify small and large at 165 without any biological criteria might be not the best approach. I understand that it was done to have a balanced groups, but even thought might be not correct. 

We were unable to determine sex for swordfish as our samples come from the commercial fisheries and we have no study on gonads at present in our institution. We would have liked to break the groups by maturity stages but it was not possible so we chose to do in this way while being aware of the problems that might arise with it.

Line 224-227: It is not clear if the statistical analysis were only used the 6 most important prey items or it was done for all the diet prey. Many times differences in diet between groups of the same species are driven by not the main prey, so I recommend to include as many prey as possible in the statistical analysis. 

The rationale for choice of prey in the statistical analysis was based on statistical power: for infrequently occurring prey, the small number of non-zero records means that statistical modelling will be uninformative. One could of course argue about the precise cut-off point but we chose to analyze prey types which occurred most frequently in our samples. A possible backup is to carry out a multivariate analysis including explanatory variables (e.g. RDA) but here too infrequently occurring prey are uninformative and indeed the signal to noise ratio in such analyses tends to be very low for dietary data.

Line 224-225: After reading the results I realized that each of the factors Size, Area, Year is analyzed separately/independently, one by one. Please, specify it in the text. 

We added the suggested text.

Line 237: Only 5 of the seven explanatory variables are mentioned (Area, year, half-year, predator size, SST). Which are the two remaining ones? Also justify why you the half year variable and which is the criteria to separate up to November 7. In line 218 -2019 you named the variables east and west ad Within SCB and Beyond SCB, be consistent. The same for the explanatory var*/iable that in line 238 is called time-period, but then is named “Year”.

We actually only included 5 variables (7 was a typo). We added a sentence that specifies what “half-year” is. As suggested, we revised ‘east and west’ to ‘Within the SCB and beyond the SCB’ to improve consistency. Time period was substituted with ‘year’.

Line 240: If you group years the objective is to have a higher number of samples for each category, but you are not reducing the number of explanatory variables, since the variable “year” is still in the analysis. The text should be corrected. 

We wrote: ”Years were grouped to avoid an excessive number of explanatory variables in relation to the sample size and to retain reasonable sample sizes per group”. For clarification of our point, we propose to revise this sentence as follows: “Years were grouped to reduce the number of distinct levels of the ‘years’ variable relative to the sample size and to retain a reasonable number of observations per year grouping. This approach concentrates more observations on each distinct level of the year variable, potentially increasing the reliability of our inferences about year.” 

Line 236-237: The effects on what? 

“On the diet as prey numbers (N)”, as added to the text.

Line 247: Which are the variables included in the GAM analysis? Which of the variables were included as continuous variables and which as factors? Did you include the half year nested within Year? 

The explanatory variables were the same used for RDA (continuous: EFL, year and SST; factors: area and half-year). We added this sentence to the text: “Half-year is a stand-alone binary variable which is not nested within year.”

Did you performed a diagnosis of the model assumptions verifying the independence of the residuals for the theoretical assumption of normality, homoscedasticity and independence? Please, include the residual plots as supplementary material and a table with the model tested and the corresponding AIC and R2. 

Because our data represent small counts of individual species found in each stomach, we don’t expect our data to follow the normal distribution and did not assume normally distributed residuals. We added the residual plots to the supplementary materials, which visually confirm the positive skewness / non-normality of the data and do not suggest the presence of heteroscedasticity. We used the negative binomial model, which may be interpreted as a generalization of the Poisson distribution that does not assume the variance equals the mean, as an appropriate model for our count data. We included explanatory variables to account for known dependencies. We also added the adjusted R2 to the GAM results table that was already present in the manuscript.

Results

Figure 1: The Map should be modified including more information. In the map indicates the name of the pacific Ocean, the countries, or some geographical details that somebody who is not familiar with the area can easily interpreted. In the figure caption you mention “Baja California) but is not obvious where it is. Describe what it means the vertical line, and also describe in the figure, and in the figure caption the what it means the scale (Nº of individuals). In the figure caption include what it means CCLME, since each figure should be self-explicative without the need of reading the manuscript. 

We added country names to the map and we specified better what the line stands for, number of individuals and we spelled out CCLME. The map was created by the authors in R. It is not a copyrighted image from another source.

Lines 273-277: Connected with my previous comment, with figure 2 it is clearly stated that for females the group below 165cm includes mature and immature individuals. Then it is not justify to separate in two groups with the criteria of having the same number of samples. 

We lack the capability to sex the individuals. Our way of separating the sample was the best approach we could come up with, given our available resources.

Table 2: The information “A total of 299 stomach was examined” is irrelevant here, mention that the table 2 correspond to the 292 individuals. 

We addressed this comment in the Table 2 caption.

Comment: Can you provide a table with the number of samples (stomachs) included per year, area, and size group? 

We added a table with the requested information (see new Table 1).

Lines 347-354: Describe also that SST was not significant. 

We added a sentence to describe results for SST.

Line 365: The criteria of AIC and Deviance explained to select models should be explained in Material and methods. Also A supplementary table with the AIC of and DE of the background selection procedure to justify the model selection should be provided. 

We inserted a paragraph explaining the use of Akaike Information Criterion (AIC) and Deviance Explained (DE) as alternative model selection criteria for GAMs. We also added an explanation of our model selection procedure of choosing the model with the lowest AIC. We believe including a supplementary table with the background selection procedure for the model would add a lot of information with limited utility to a paper that is already quite long, so have not undertaken this additional step. We do provide a table with the model parameters that we chose based on lowest AIC, which we feel is the relevant information the reader needs to understand the results of our model selection procedure. 

Line 365: Please, could you better explain in material and methods and results which data are you using in the GAM models as response variable? As it is right now is confusing. You run the models at individual stomach level? But then I don’t understand if it is presence absence, like stated in line 367, why it is said that was ranked by GII, but in Material and methods is indicated that was count data (number of preys items in each stomach). 

The analysis was conducted for all stomachs in the sample at once, not at the individual stomach level. The response variable was the number of prey items for the species in each stomach, not presence-or-absence data. The negative binomial link function was used to analyze the count data for number of prey items in each stomach. We have corrected the description in the Results to clarify the approach. 

Figure 4. If is a single figure with several panels, it should appear as a single figure. Now is split in 7 different figures. Also, in the figure caption it should be described to which species correspond each letter. 

We built 2 figures (Figure 4 and 5) to consolidate the 7 figures. We updated the text and the caption.

In the results I am missing the results from genetic identification. It would be interesting to include which were the species that thanks to including DNA analysis could be incorporated to understand the bias that is done in the description of diets when not including this type of identification. 

The DNA analysis was performed only on three prey specimens, two chubby pearleye and a luvar. We added a clarifying sentence in the results.

Discussion:

From line 405 to 427 the text should be removed since is information not supported with your results. If some of the information provided in this two paragraphs you think is important to be mentioned, it can be moved to Introduction. The Discussion should start in line 428. 

We removed lines 405 to 427 as suggested.

Line 459: The range of prey size in material an methods was indicated that it was measured, but then in results I couldn’t find anything reporting this results. It would be very interesting to have this information available. 

We added Table 2 with range, mean and median prey size. We inserted a sentence in data analysis: “Size range for prey in fresh and intermediate state of digestion was reported by species. Mean and median prey size was calculated for prey species with at least 2 specimens” plus one sentence in results: “Prey size was measured for 328 specimens of 22 prey species in a fresh state of digestion. Prey size range was reported and mean and median prey size by species were calculated for prey with at least 2 specimens available.”

The discussion in general is well written and interesting, but the results from GAM regarding the variables SST and Half-year are not discussed.

We inserted a paragraph on the effects of temperature on the behavior of jumbo squid and added a small paragraph in the discussion regarding the influence of Half-year for Gonatus spp. and duckbill barracudina. 

I know that the diet is already reported in tables, but a graph with the diet by year period would facilitated the intercomparison of changes in diet between years to better follow the discussion. But it is up to the authors to included or not, since I understand that this information is already provided. 

We added a color coded barplot (Figure 4) that illustrates the % number of prey by year.

Comments from Reviewer #2

Even so, I think the description of the fishery from page 2, line 51 to page 3, line 60 could probably be simplified. 

Following the suggestion, we removed the following sentence “Swordfish command a high economic value in both commercial and recreational fisheries in all oceans of the world [4].” 

We also removed this sentence: “The swordfish population in the North Pacific is assessed as two stocks, divided by a boundary extending from Baja California (25ºN x 110ºW) to 165ºW at the Equator [10, 11]. These are the Western and Central North Pacific Ocean (WCNPO) stock and the Eastern Pacific Ocean (EPO) stock [3, 12, 13]. The most recent stock assessment indicated that the WCNPO stock, which is the source of the DGN fleet swordfish catch, was neither overfished nor experiencing overfishing [13].”

In addition, page 4, lines 91-95 and Table 1 should be moved to Discussion to consider the geographic variation in swordfish diet. 

Text and table were moved to the discussion as suggested. 

I agree, that GII and IRI are useful to get an overview of the importance of prey species. However, I think each of the three RMPQs used in this calculation has a different meaning. For example, F would reflect foraging opportunities. In the case of a predator that picks up individual prey items, such as swordfish, N would reflect an aspect of prey availablity and foraging effort; W would relate to its importance as an energy source. Therefore, I think it is necessary to consider the RMPQs of each major prey species separately. For example, Abraliopsis, a small squid, is not an important prey in terms of weight, although its high F and N indicate that it is fed frequently. Nevertheless, it is important to note that GII and IRI are high. As an opposite pattern, luvar has a large body size, and even though the weight index is large, both F and N are low, suggesting that its importance as food is limited to a few individuals. 

This is a very good observation. It is true that each of the three RMPQs used in this calculation has a different meaning. We added a sentence in the discussion that reports this information. 

In this manuscript, the first part of the discussion discusses the overall characteristics of the species that swordfish use as their primary food item based on GII. It concludes that the swordfish will have a flexible choice of medium or large prey items related to DSL. I think this is a reasonable conclusion. However, I think it would be desirable to be a little more careful in the discussions that lead to this conclusion. For example, with regard to the size of the prey species, it would be good to consider the characteristics of the main prey species in the list of prey species that were actually preyed upon, and come to a conclusion as to what size range the swordfish considers to be its main prey. In the current manuscript, this is only briefly described on page 23, lines 444 to 448. 

We added a sentence that indicates the average sizes of the main prey items and the smallest ones, like market squid, to put this topic in perspective.

Similarly, I think the authors should distinguish between DSL-related mesopelagic prey and other epipelagic prey on the list before concluding on foraging depth. This careful treatment is especially necessary for PlosOne readers. As shown in Figure 3, I believe that present study has achieved a near upper limit of prey variation, although it is not perfect. Therefore, I think that such an evaluation of the diet overview is important in describing the general dietary characteristics of swordfish.

We specified that “A number of the most important swordfish prey species are found in or associated with the DSL, including jumbo squid, G. borealis and Gonatus spp. squids, barracudinas, and Pacific hake. Other important prey, like Abraliopsis sp. and market squid, are more epipelagic. The range of prey species eaten, in terms of both prey size and prey habitat, suggests that swordfish have quite flexible foraging strategies”. 

Also, on page 4, line 85 of the preface, the authors describe the need for energy-rich food, but have you seen any general trend toward that? 

We removed the sentence “Thus, they need to catch more energy-rich prey or consume a greater quantity of prey than would be necessary if they were ectothermic” since we are not sure how to determine a defined trend. Cephalopods are not the most energetic foods.

Other miscellaneous points;

P8 lines140-143: What is the reference to identify fish otolith and squid beaks? Please specify literature or reference sample collection.

 We added the requested references.

P10 lines 193-194: Empty stomachs were excluded from contribution analysis, but empty stomachs should be included in the calculation of frequency of occurrence. 

Empty stomachs can either be included or excluded when estimating frequency of occurrence; different authors use different methods. We base our analyses on Hyslop, E.J., 1980. Stomach contents analysis—a review of methods and their application. Journal of fish biology, 17(4), pp.411-429. He states in his publication: “Possibly the simplest way of recording data gleaned from stomach contents is to record the number of stomachs containing one or more individuals of each food category. This number may then be expressed as a percentage of all stomachs (Frost, 1946, 1954; Hunt & Carbine, 1951) or all those containing food (Dineen, 1951; Dunn, 1954; Kennedy & Fitzmaurice, 1972).”

P14 lines 278-279: 91% of the food items were severely digested. To what extent is this expected to affect the weight composition of the prey? The extent to which it affects the assessment of importance by GII should be added to the discussion.

It is important to consider that these results on importance of prey are based on GII and IRI calculated with 91% of prey that were in an advanced state of digestion. If prey had been in a more recent state of digestions results could have been different. It is understood that using weight of prey remains can bring a good amount of bias. We added a sentence in the discussion to acknowledge this point.

P19 Table 3: SST was not significant factor in the results of RDA, but in P20 lines 368-369, SST was included final model of GAM for jumbo squid. How can this inconsistency be interpreted?

This may reflect the difference between using RDA, which is a multivariate method, versus applying the GAM model to individual species stomach count data. In general, different methods will select different variables for inclusion in the best model. For example, we included sea surface temperature as an explanatory variable in RDA, but it did not come out as significant. In general, a variable that is significant in a univariate analysis might not be in a multivariate analysis.

Table S15: Y1 = S1? 

It was a typo from a previous version. We have corrected this in the manuscript.

Comments from Reviewer #3

Abstract

Lines 26-27. Remove “… federal….boats…” 

We removed ‘federal’ but we feel “boats” is needed. 

Line 35. Remove “…in swordfish diet…” 

We removed the suggested words. 

Line 36. Change “form” to “in” 

We have made this change.

Line 37. Change “with Pacific hake the…” to “Pacific hake being”. 

We have made this change.

Line 40. Change “from to “during”. 

We have made this change.

Line 41. Which factors? Please clarify. 

We added “(swordfish size, area, time period, sea surface temperature)”.

Line241-42. Standarditation? Please clarify. 

We changed to “standardizing methods”.

Introduction

Lines 48-50. “Sowrdfish… dolphins” Please rewrite. 

We rewrote the sentence.

Line 50. Change “command” to “have”. 

We removed the sentence per another reviewer’s suggestion.

Lines 53-54. “Swordfish…fishery”. Please rewrite. 

We rewrote the sentence in a less repetitious way.

Line 64. Horizontal and vertical movements of the swordfish in the southern Pacific Ocean have been studied by Abascal et al (2010) and Evans et al (2014).

References:

- Abascal FJ, Mejuto J, Quintans M, Ramos-Cartelle A (2010) Horizontal and vertical movements of swordfish in the Southeast Pacific. ICES J Mar Sci 67:466–474

- Evans K, Abascal F, Kolody D, Sippel T, Holdsworth J, Maru P (2014) The horizontal and vertical dynamics of swordfish in the South Pacific Ocean. J Exp Mar Biol Ecol 450:55–67 

Although there is no evidence of trans-equatorial or trans-Pacific crossing, data suggest that SCB swordfish may exhibit a higher level of Eastern Pacific Ocean (EPO) connectivity than previously proposed; we added a sentence to clarify this point. We also cited the suggested references.

Line 77. Swordfish can reach depths of up to 1136m (Abascal et al, 2010). 

We added this reference and corrected the depth mentioned in the manuscript to this value.

Lines “106-110”. According to the authors, there are 5 previous studies aimed to investigate the feeding habits of the swordfish in the area. Which is the novelty of the present study?

We added two paragraphs at the end of the introduction to explain the novelty of the study and to motivate some potential applications. 

In my opinion, the authors should provide information on the consumption rate using the model proposed by Olson and Mullen (1986). See also Olson and Boggs (1986) and Olson and Galván-Magaña (2002). References:

- Olson RJ, Boggs CH (1986) Apex predation by yellowfin tuna (Thunnus albacares): independent estimates from gastric evacuation and stomach contents, bioenergetics, and cesium concentrations. Can J Fish Aquat Sci 43:1760–1775

- Olson RJ, Galván-Magaña F (2002) Food habits and consumption rates of common dolphinfish (Coryphaena hippurus) in the eastern Pacific Ocean. Fish Bull 100:279–298

- Olson RJ, Mullen AJ (1986) Recent developments for making gastric evacuation and daily ration determinations. Environ Biol Fish 16: 183–191 

We reviewed the recommended references and considered the additional work that would be needed to compute feeding consumption rates. We lack data for the study area on the average amount of time required to evacuate the average proportion of all meals present in the stomach at any instant in time, which is a component of the formula. While we recognize the potential value of these results, we are also concerned that including them would go beyond the scope of the paper. We have continued stomach contents data collection since 2014, and will consider computing feeding consumption rates as part of a future project.

Table 1. The following studies have not included in this table.

- Abid N, laglaoui A, Arakrak A, Bakkali M (2018) The role of fish in the diet of swordfish (Xiphias gladius) in the Strait of Gibraltar. J Mar Biol Assoc UK 4: 895-907. 

We added this reference for the western Mediterranean Area.

- Holts D, Sosa-Nishizaki O (1998) Swordfish, Xiphias gladius, fisheries of the eastern North Pacific Ocean. In: Barret I, Sosa-Nishizaki O, Bartoo N (eds) Biology and fisheries of swordfish, Xiphias gladius. Papers from the International Symposium on Pacific Swordfish, Ensenada, Mexico, 11–14 December 1994. US Department of Commerce, NOAATechnical Report NMFS 142, pp 65–76. 

The suggested paper (Holts D, Sosa-Nishizaki O (1998)) is not a diet study. We already included a diet study referenced in this technical report: “Markaida U, Sosa-Nishizaki O. Food and feeding habits of swordfish, Xiphias gladius L, off western Baja California. Biology and fisheries of Swordfish, Xiphias gladius. NOAA Tech. Rep. 1998; 142 p 245-259”.

Additionally, we added the following three publications to our references.

- Logan JM, Golet W, Smith SC, Neilson J, Van Guelpen L (2021) Broadbill swordfish (Xiphias gladius) foraging and vertical movements in the north-west Atlantic. J Fish Biol 99: 557-568.

- Young JW, Lansdell MJ, Campbell RA, Cooper SP, Juanes F, Guest MA (2010) Feeding ecology and niche segregation in oceanic top predators off eastern Australia. Mar Biol 157:2347–2368. 

- Zambrano-Zambrano RW, Mendoza-Moreira PE, Gómez-Zamora W, Varela JL (2019) Feeding ecology and consumption rate of broadbill swordfish (Xiphias gladius) in Ecuadorian waters. Mar Biodiver 49:373-380.

Material and methods

Line 141. Which taxonomic keys? 

Taxonomic keys references were added.

Lines 145-150. In a recent review article, Amundsen and Sánchez-Hernández (2019) have criticized the fact of estimating prey weight from measurements of hard parts. Please, include it in Discussion section. --Amundsen P-A, Sánchez-Hernández J (2019) Feeding studies take guts – critical review and recommendations of methods for stomach contents analysis in fish. J Fish Biol 95:1364-1373. 

We added a paragraph in the discussion section to reflect the Amundsen and Sánchez-Hernández concern about possible bias due to estimating prey weight from measurement of hard parts. 

Line 197. According to Brown et al (2013), PSIRI provide more accurate estimates than IRI. Please, calculate PSIRI. 

We calculated PSIRI and added a section in methods. We also updated the related tables in the supplemental materials.

Provide information on how feeding consumption rate has been estimated. According to Olson and Galván-Magaña (2002) weight data estimated form hard parts (cephalopod beaks and/or fish otoliths) should not be considered in this analysis.

Reference:

- Olson RJ, Galván-Magaña F (2002) Food habits and consumption rates of common dolphinfish (Coryphaena hippurus) in the eastern Pacific Ocean. Fish Bull 100:279–298

As noted above, we reviewed the recommended references and considered the additional work that would be needed to compute feeding consumption rates. We lack data for the study area on the average amount of time required to evacuate the average proportion of all meals present in the stomach at any instant in time, which is a component of the formula. While we recognize the potential value of these results, we are also concerned that including them would go beyond the scope of the paper. We have continued stomach contents data collection since 2014, and will consider computing feeding consumption rates as part of a future project.

Line 227. Why did you only use GII values for the analysis? Please explain. 

The ranks of GII and IRI are very similar. It was done out of simplicity. We stated it at line 303 “Rankings of prey taxa based on GII and IRI were nearly identical”.

Results

Provide consumption rate data by size class and area. 

See explanation above: We reviewed the recommended references and considered the additional work that would be needed to compute feeding consumption rates. We will consider computing feeding consumption rates as part of a future project

Lines 274. Change “[109]” to “authors et al [109]. 

We addressed this point.

Lines 274-277 “[109]… determined”. Remove it. As the authors have stated, they did not record fish sex. 

While we appreciate the reviewer’s concern, we feel we need to keep this sentence since it explains the potential maturity level of our samples. 

Lines 291-296. Considering the p value reported by the authors, it looks like the number the samples were not enough to describe the diet completely. In fact, the authors stated in Discussion section that this study “would have benefited from a large sample size since” the curve did not reach the asymptote. Please, indicate how the low number the samples may affect the robustness of your results. 

We added a sentence in the discussion explaining that the cumulative curve covered the most important prey items in the swordfish diet, but we may be missing some the less frequently encountered ones. In contrast to some other research that we cite, we identified a large majority of prey items to the species level, making the asymptote more difficult to reach.

Lines 302-313. Please provide the values of the alimentary indices (when possible).

We added the values %GII and %IRI by the prey.

Lines 326-345. Why did you only compare GI values, but no IRI (or PSRI)? Clarify.

We took this approach because, as we stated at line 303, the ranks of GII and IRI are very similar. We believe that additionally reporting IRI could be redundant. We stated “Rankings of prey taxa based on GII and IRI were nearly identical”.

A table including number the samples collected by area, size class and year is missed. In this table, the authors should also include the number of samples with at least one prey by group (and %percentage of non-empty stomachs). 

We added a Table (now Table 1) with the information required.

Discussion

Lines 414-417. “Adult…whales” Remove it. This information is not in line with the manuscript. The information was removed, as requested.

Line 444. According to Gilly et al (2006), the jumbo squid is a mesopelagic cephalopod. Please

check the following MS: - Gilly W.F., Markaida U., Baxter C.H., Block B.A., Boustany A., Zeidberg L., Reisenbichler K., Robison B., Bazzino G. and Salinas C. (2006) Vertical and horizontal migrations by the jumbo squid Dosidicus gigas revealed by electronic tagging. Marine Ecology Progress Series 324, 1–17. 

We modified the text and added the recommended reference.

---

## [Decision Letter · Decision Letter 1]

25 Jul 2022

PONE-D-21-29802R1Feeding ecology of broadbill swordfish (Xiphias gladius) in the California CurrentPLOS ONE

Dear Dr. Preti,

Thank you for submitting your manuscript to PLOS ONE. The three reviewers who originally reviewed this paper have also considered the versions. Your paper is nearly ready to accept, pending some minor changes and clarifications. Well done on the revisions. Therefore, we invite you to submit a revised version of the manuscript that addresses the points raised during the review process.

We look forward to receiving your revised manuscript.

Kind regards,

Antonio Medina Guerrero, Ph.D.

Academic Editor

PLOS ONE

Journal Requirements:

Reviewers' comments:

Reviewer's Responses to Questions

**Comments to the Author**

1. If the authors have adequately addressed your comments raised in a previous round of review and you feel that this manuscript is now acceptable for publication, you may indicate that here to bypass the “Comments to the Author” section, enter your conflict of interest statement in the “Confidential to Editor” section, and submit your "Accept" recommendation.

Reviewer #1: All comments have been addressed

Reviewer #2: All comments have been addressed

Reviewer #3: All comments have been addressed

2. Is the manuscript technically sound, and do the data support the conclusions?

Reviewer #1: Yes

Reviewer #2: Yes

Reviewer #3: Yes

3. Has the statistical analysis been performed appropriately and rigorously? 

Reviewer #1: Yes

Reviewer #2: Yes

Reviewer #3: Yes

4. Have the authors made all data underlying the findings in their manuscript fully available?

Reviewer #1: Yes

Reviewer #2: Yes

Reviewer #3: Yes

5. Is the manuscript presented in an intelligible fashion and written in standard English?

Reviewer #1: Yes

Reviewer #2: Yes

Reviewer #3: Yes

6. Review Comments to the Author

Reviewer #1: The manuscript has been much improved and all the comments were answered and addressed by the authors. I really appreciate the effort of the authors in the response document, all the modifications that have been done are well explained. The novelty of the study is now clearly stated. Then, I consider that, after some minor considerations (see comments below), the manuscript is ready for publication.

Minor comments

Introduction:

Line 49: What do you mean with “they are productive”? Perhaps you wanted to say that they are highly abundant or that they are meso-predators?

Line 99-119: I appreciate the explanation that authors have included to answer my comment of being more concrete on how their findings can be used. The text is highly complete, but very long. Perhaps you could summarize the text with something similar to:

Due to the complexity of many ecosystems, there is a need for basic knowledge of trophic interactions that are critical to understand system productivity and food chain dynamics. New policy developments have increased the relevance of feeding ecology studies, as policy-makers and fisheries managers have embraced the concept of ecosystem-based fisheries management (EBFM), thus taking a more holistic approach to resource management [34, 35]. The findings of this study can serve to inform ecosystem models considering trophic interactions and contribute to the development of alternative approaches to better manage this economically and ecologically important species and move towards an EBFM.

Material and Methods:

The lack of ecological criteria to divide large and small individuals and to divide the half year in two equal periods, in my opinion is not the best option, but considering that how the analysis was conducted is well explained and the authors have justified the use of this criteria, I will not insist in that aspect.

Lines 291-294: If DE is not used for model selection, it is not necessary to give all of this explanation on the use of DE, and these sentences can be removed.

Lines 230-231: Since in the discussion you relate differences in diet with the characteristics of the sampling area (within or beyond SCB), please give some information/description of the difference between the SCB area and the rest of the sampling area. For example, include if SCB is a shallower area or influenced with specific currents, and if SCB is considered more inshore than the rest as mentioned in the discussion (lines 565-568). For somebody not familiar with the area of study, SCB sampling points do not look more inshore than the rest of the sampling points.

Results:

Line 404: The description of the legend (Red= Teuthoidea; Blue = Teleostei…) is already described in the figure 4 caption, therefore it can be deleted from the body text.

Line 447: Thank you for including the R-adj sq values. Considering these values, I wanted to add an additional comment. In the GAM model for Abraliopsis sp., the R-sq (adj.) is negative indicating that the model is not good or better than a straight line. Then, this model should not be considered as valid.

Discussion:

Thank you for addressing my comments, but still I don’t see the connection of the paragraph 480-489 with your results, please can you connect this first paragraph with the discussion of your results?

Reviewer #2: (No Response)

Reviewer #3: (No Response)

7. PLOS authors have the option to publish the peer review history of their article (what does this mean?). If published, this will include your full peer review and any attached files.

Reviewer #1: No

Reviewer #2: **Yes: **Hiroshi Ohizumi

Reviewer #3: No

---

## [Author Response · Author response to Decision Letter 1]

15 Sep 2022

Dr. Antonio Medina Guerrero

Academic Editor

PLOS ONE

Dear Dr. Medina Guerrero, 

Thank you for the opportunity to submit a second revised draft of our manuscript titled: “Feeding ecology of broadbill swordfish (Xiphias gladius) in the California Current”. Once again we appreciate the time and effort that you and the reviewers have dedicated to providing your valuable feedback and insightful comments to our manuscript. We have incorporated the new changes to reflect the suggestions reviewers provided in the best way we could. We have highlighted our proposed changes to the manuscript through MS Word’s ‘track changes’ feature. Here is a point-by-point response to the reviewers’ new comments and concerns.

Thank you so much for your help and your time.

Sincerely,

Antonella Preti and coauthors

Minor comments

Introduction:

Line 49: What do you mean with “they are productive”? Perhaps you wanted to say that they are highly abundant or that they are meso-predators?

We decided to remove the statement as it is not central to the topic.

Line 99-119: I appreciate the explanation that authors have included to answer my comment of being more concrete on how their findings can be used. The text is highly complete, but very long. Perhaps you could summarize the text with something similar to:

Due to the complexity of many ecosystems, there is a need for basic knowledge of trophic interactions that are critical to understand system productivity and food chain dynamics. New policy developments have increased the relevance of feeding ecology studies, as policy-makers and fisheries managers have embraced the concept of ecosystem-based fisheries management (EBFM), thus taking a more holistic approach to resource management [34, 35]. The findings of this study can serve to inform ecosystem models considering trophic interactions and contribute to the development of alternative approaches to better manage this economically and ecologically important species and move towards an EBFM.

We thank the reviewer for the detailed explanation and help. In line with the suggestion, we updated this section as follows:

“Due to the complexity of many ecosystems, there is a need for basic knowledge of trophic interactions that are critical to understand system productivity and food chain dynamics. New policy developments have increased the relevance of feeding ecology studies, as policy-makers and fisheries managers have embraced the concept of ecosystem-based fisheries management (EBFM), thus taking a more holistic approach to resource management [34, 35]. The findings of this study can inform ecosystem models with information about trophic interactions, contributing to the development of alternative approaches to better manage this economically and ecologically important species.” 

Material and Methods:

The lack of ecological criteria to divide large and small individuals and to divide the half year in two equal periods, in my opinion is not the best option, but considering that how the analysis was conducted is well explained and the authors have justified the use of this criteria, I will not insist in that aspect.

We thank the reviewer for sharing their perspective.

Lines 291-294: If DE is not used for model selection, it is not necessary to give all of this explanation on the use of DE, and these sentences can be removed.

We removed the sentences as suggested, and slightly modified the subsequent sentence to read, “The AIC trades off higher values of the likelihood function against a penalty for adding more parameters.”

Lines 230-231: Since in the discussion you relate differences in diet with the characteristics of the sampling area (within or beyond SCB), please give some information/description of the difference between the SCB area and the rest of the sampling area. For example, include if SCB is a shallower area or influenced with specific currents, and if SCB is considered more inshore than the rest as mentioned in the discussion (lines 565-568). For somebody not familiar with the area of study, SCB sampling points do not look more inshore than the rest of the sampling points.

We revised the language to clarify the difference between the SCB area and currents compared to the rest of the sampling area, as follows: “…‘within the SCB’ (east of 120º 30’W longitude) and ‘beyond the SCB’ (west of 120º 30’W longitude), reflecting separation between the more inshore waters in the SCB where the northward flowing California Counter Current influences nearshore oceanography and the more offshore waters affected by the California Current as it moves southward…”

Results:

Line 404: The description of the legend (Red= Teuthoidea; Blue = Teleostei…) is already described in the figure 4 caption, therefore it can be deleted from the body text.

We deleted the text as suggested.

Line 447: Thank you for including the R-adj sq values. Considering these values, I wanted to add an additional comment. In the GAM model for Abraliopsis sp., the R-sq (adj.) is negative indicating that the model is not good or better than a straight line. Then, this model should not be considered as valid.

While we concede the reviewer’s point that the GAM model for Abraliopsis sp. did not explain much of the variation in swordfish diet, we nonetheless include it in our results for completeness. We interpret the negative R-sq (adj.) to indicate a very weak or nonexistent relationship between Abraliopsis sp. consumption and the explanatory variables in our model. We added a statement in the manuscript that the Abraliopsis sp. model was unsatisfactory as indicated by the negative R-square adjusted.

Discussion:

Thank you for addressing my comments, but still I don’t see the connection of the paragraph 480-489 with your results, please can you connect this first paragraph with the discussion of your results?

To make this connection, we added the following sentence at the beginning of the paragraph: “The range of prey species found in our study is consistent with the diurnal vertical distribution of swordfish, reflecting their diving behavior.”

Reviewer #2: (No Response)

Reviewer #3: (No Response)

7. PLOS authors have the option to publish the peer review history of their article. If published, this will include your full peer review and any attached files.

No. We would prefer not to have this option.

Do you want your identity to be public for this peer review? For information about this choice, including consent withdrawal, please see our Privacy Policy.

Reviewer #1: No

Reviewer #2: Yes: Hiroshi Ohizumi

Reviewer #3: No

We have added mention of Hiroshi Ohizumi’s contribution as a reviewer to our acknowledgments.

---

## [Decision Letter · Decision Letter 2]

18 Oct 2022

Feeding ecology of broadbill swordfish (Xiphias gladius) in the California Current

PONE-D-21-29802R2

Dear Dr. Preti,

We’re pleased to inform you that your manuscript has been judged scientifically suitable for publication and will be formally accepted for publication once it meets all outstanding technical requirements.

Kind regards,

Antonio Medina Guerrero, Ph.D.

Academic Editor

PLOS ONE

Additional Editor Comments (optional):

Reviewers' comments:

Reviewer's Responses to Questions

**Comments to the Author**

1. If the authors have adequately addressed your comments raised in a previous round of review and you feel that this manuscript is now acceptable for publication, you may indicate that here to bypass the “Comments to the Author” section, enter your conflict of interest statement in the “Confidential to Editor” section, and submit your "Accept" recommendation.

Reviewer #1: All comments have been addressed

2. Is the manuscript technically sound, and do the data support the conclusions?

Reviewer #1: Yes

3. Has the statistical analysis been performed appropriately and rigorously? 

Reviewer #1: Yes

4. Have the authors made all data underlying the findings in their manuscript fully available?

Reviewer #1: No

5. Is the manuscript presented in an intelligible fashion and written in standard English?

Reviewer #1: Yes

6. Review Comments to the Author

Reviewer #1: The revised manuscript has been improved and all the minor comments of this second review have been answered and modified accordingly. I do not have any firther comments to add. Then, I consider that the manuscript is ready for publication.

7. PLOS authors have the option to publish the peer review history of their article (what does this mean?). If published, this will include your full peer review and any attached files.

Reviewer #1: No

---

## [Editor Report · Acceptance letter]

8 Dec 2022

PONE-D-21-29802R2 

 Feeding ecology of broadbill swordfish (*Xiphias gladius*) in the California Current 

Dear Dr. Preti:

I'm pleased to inform you that your manuscript has been deemed suitable for publication in PLOS ONE. Congratulations! Your manuscript is now with our production department. 

Kind regards, 

on behalf of

Dr. Antonio Medina Guerrero 

Academic Editor

PLOS ONE